# Assessment of the Biocompatibility Ability and Differentiation Capacity of Mesenchymal Stem Cells on Biopolymer/Gold Nanocomposites

**DOI:** 10.3390/ijms25137241

**Published:** 2024-06-30

**Authors:** Huey-Shan Hung, Chiung-Chyi Shen, Jyun-Ting Wu, Chun-Yu Yueh, Meng-Yin Yang, Yi-Chin Yang, Wen-Yu Cheng

**Affiliations:** 1Graduate Institute of Biomedical Science, China Medical University, Taichung 404328, Taiwanhs0603@gmail.com (J.-T.W.); 2Translational Medicine Research, China Medical University Hospital, Taichung 404327, Taiwan; 3Department of Minimally Invasive Skull Base Neurosurgery, Neurological Institute, Taichung Veterans General Hospital, Taichung 407204, Taiwan; ccshen@vghtc.gov.tw (C.-C.S.); jean1007@gmail.com (Y.-C.Y.); 4School of Medicine, China Medical University, Taichung 404333, Taiwan; 5Department of Post-Baccalaureate Medicine, College of Medicine, National Chung Hsing University, Taichung 402202, Taiwan; 6Institute of Biomedical Sciences, National Chung Hsing University, Taichung 402202, Taiwan; 7Taiwan Department of Physical Therapy, Hung Kuang University, Taichung 433304, Taiwan

**Keywords:** Fibronectin, type I collagen, gold nanoparticles, mesenchymal stem cells, biocompatibility, multiple differentiation, surface characteristic, animal models

## Abstract

This study assessed the biocompatibility of two types of nanogold composites: fibronectin-gold (FN-Au) and collagen-gold (Col-Au). It consisted of three main parts: surface characterization, in vitro biocompatibility assessments, and animal models. To determine the structural and functional differences between the materials used in this study, atomic force microscopy, Fourier-transform infrared spectroscopy, and ultraviolet-visible spectrophotometry were used to investigate their surface topography and functional groups. The F-actin staining, proliferation, migration, reactive oxygen species generation, platelet activation, and monocyte activation of mesenchymal stem cells (MSCs) cultured on the FN-Au and Col-Au nanocomposites were investigated to determine their biological and cellular behaviors. Additionally, animal biocompatibility experiments measured capsule formation and collagen deposition in female Sprague–Dawley rats. The results showed that MSCs responded better on the FN-Au and Col-AU nanocomposites than on the control (tissue culture polystyrene) or pure substances, attributed to their incorporation of an optimal Au concentration (12.2 ppm), which induced significant surface morphological changes, nano topography cues, and better biocompatibility. Moreover, neuronal, endothelial, bone, and adipose tissues demonstrated better differentiation ability on the FN-Au and Col-Au nanocomposites. Nanocomposites have a crucial role in tissue engineering and even vascular grafts. Finally, MSCs were demonstrated to effectively enhance the stability of the endothelial structure, indicating that they can be applied as promising alternatives to clinics in the future.

## 1. Introduction

Materials less than 100 nm in length are referred to as nanomaterials. They can be classified into four types: zero-dimensional (nanoparticles) [1], one-dimensional (nanotubes) [2], two-dimensional (nanofilms) [3], and three-dimensional (nano-bulk materials) [4]. Nanomaterials have unique characteristics, such as their small size, high surface area, and quantum tunneling effect. They have various applications in pharmacology and biological therapy as biomaterials, which can be used for diagnosis and treatment. For example, a nanotechnology-based thin film can be used as a vascular graft by modifying the size of microporous structures, which can prevent inflammatory cell penetration. Moreover, nanomaterials can be used as drug carriers to improve local endothelialization and drug penetration, preventing vessel recocclusion [5].

Since gold nanoparticles (AuNPs) tend to aggregate to microscale bulk gold, they are commonly prepared in solution [6]. When suspended in a solution, AuNPs show characteristics similar to a suspension and have a sub-micron size (<1 µm) [7]. Moreover, they possess other excellent properties, such as softness, stability, inertness, high density, high extension, and, most importantly, outstanding biocompatibility [8]. AuNPs have highly efficient photothermal and photoelectric effects, making them one of the most widely studied nanomaterials. They are popular in electric fields, such as electron microscopy, electrical engineering, or electronics, and they can be applied in the medical field as vascular grafts, as drug carriers, and for image analysis, which are the goals of the modern nanomedicine field [9,10].

Previous studies found polyurethane (PU) to possess excellent elasticity and mechanical strength. AuNPs have been used to modify PU to create nanocomposites, which have been found to be a suitable choice for vascular tissue engineering due to their anti-thrombus properties [11]. Our group has conducted experiments using various AuNP concentrations (17.4–174.0 ppm) and discovered that 43.5 ppm resulted in the best cell migration rate [12]. Studies indicate that nanocomposites can induce the integrin αVβ3/protein tyrosine kinase 2 (PTK2/FAK) pathway to promote the proliferation and migration of endothelial cells and adult fibrocytes. It has also been observed that adult fibrocytes have a greater effect on endothelial nitric oxide synthase (eNOS) and FAK/Rho-GTPase than endothelial cells [13].

Regarding surface modification, combining AuNPs with fibronectin (FN1/FN) can improve biocompatibility with the cardiovascular system, as evidenced by adhesion tests of monocytes and platelets. Previous research has found that using 43.5 ppm AuNPs leads to the highest rate of cell proliferation, lower generation of reactive oxygen species (ROS), and increased levels of matrix metallopeptidase 9 (MMP9) and phosphorylated (p)-eNOS. These findings indicate that nanocomposites can improve the migration of mesenchymal stem cells (MSCs). Based on these results, nanocomposites could be an excellent choice for modifying the surface of artificial blood vessels [14]. Our previous studies used both FN and type I collagen (Col). Our findings showed that incorporating AuNPs into Col improved the scaffold’s mechanical properties. The Col-Au nanocomposite can potentially enhance MSC proliferation and migration and stimulate endothelial cell growth, aiding vascularization. Additionally, our research indicates that 12.2 and 43.5 ppm AuNPs had similar effects on stem cell differentiation and migration. Therefore, we selected 12.2 ppm as the dosage for this study.

The extracellular matrix (ECM) is a significant component of animal tissues and determines the characteristics of connective tissue. The main components of the ECM include FN, elastin (ELN), laminins, and different types of collagen [15]. The receptors on the cell surface bind to the ECM, leading to cellular interactions and behaviors, such as migration, proliferation, and differentiation [16]. Additionally, the ECM stores several growth factors and cytokines that facilitate cell signal communication. Biomaterials and tissue engineering have flourished in recent years, with many studies using ECM peptides and proteins as biomaterials to design biomimetic materials that can be used in humans without requiring later removal [17]. Moreover, the ECM’s physical properties, surface charge, and morphology make it a common choice for surface modification. 

FN is a dimeric protein that can exist in different forms depending on the situation and has a molecular weight of around 500 kDa [18]. While it can dissolve in blood plasma, it becomes insoluble in the ECM. FN is crucial for cellular behaviors, such as adhesion, migration, and proliferation, since it has multiple binding sites for fibrin, integrins, heparan sulfate, and collagen [19]. It can also stimulate the synthesis of many functional proteins and aid in generating the endothelium while inhibiting coagulation [20]. Biopolymer materials used clinically must be biocompatible, stable, and sterile and have high mechanical strength to ensure favorable clinical prognoses. FN plays a crucial role in vascularization and microenvironment maintenance since it is abundant in the ECM of regenerating tissues, basement membranes, injured tissues, and modern biomaterials [21].

Collagens comprise around 20% of the total protein content in human tissues. They are primarily found in connective tissue and are biocompatible and degradable, making them essential for many physiological and biochemical functions. They are used in various fields, such as medicine and biomaterials. Col is the most common type of collagen and is a popular choice for research and tissue engineering since it contains specific peptide sequences that form triple-helix structures, which contain one glycine for every three amino acids and are stabilized by proline and hydroxyproline [22]. Collagens can recombine into other forms and are widely used in various medical fields, including hemostasis, nerve reconstruction, tissue plastic, drug release, ophthalmology, and cardiovascular engineering [23]. Col is the most used type of collagen in regenerative medicine due to its low antigenicity, cytotoxicity, and excellent biodegradability. It is also known for its ability to stimulate angiogenesis both in vitro and in vivo. However, there are concerns about its durability due to its rapid proteolytic degradation by macrophages in vivo. Various methods have been used to enhance its lifespan, including combining Col with AuNPs to improve its stability and reduce its degradation [24].

Integrins are key molecules that play a crucial role in cell adhesion and vasculogenesis by acting as a receptor via which cells can attach to the ECM [25]. They are dimeric protein complexes comprised of an α-chain and β-chain [26]. Various integrins, including α1β1, α2β1, α10β1, α11β1, and αVβ1, contribute to cell attachment [27]. The ECM contains various amino acid sequences that can bind to particular receptors. One such sequence is arginine-glycine-aspartate (Arg-Gly-Asp or RGD), the binding site for integrins [28]. Integrins can form complexes with various ECM proteins, including collagens, FN, and laminins. Some integrins are associated with angiogenesis, with α2, α4, αVβ1, and αVβ3 interacting with the ECM to directly regulate cell differentiation, proliferation, and migration [29]. ECM components can significantly influence the behavior of endothelial cells during angiogenesis [30], with collagens, laminins, and FN able to induce endothelial cells to adhere to the ECM surface, thereby stimulating angiogenesis. This discovery has been extremely beneficial in the medical field, with FN and collagens deemed promising biomaterials for repairing tissue damage [31].

Umbilical cord MSCs (UCMSCs) are primarily found in the human umbilical cord. However, they can also be obtained from various other tissues, such as fat, peripheral blood, skin, fetal blood, fetal bone tissue, lungs, and umbilical cord blood. UCMSCs can self-renew and differentiate into multiple cell types, making them useful for bone tissue engineering [32]. Depending on the conditions, MSCs from different tissues can be induced to become bone, endothelial, fat, or nerve cells [33]. Stem cells are a valuable biomaterial in regenerative medicine since they can transform into various tissues, including bone and nerves. Autologous transplantation eliminates the risk of immune rejection. Using scaffold materials and signaling factors in tissue engineering can aid stem cell differentiation and create a 2D or 3D tissue structure, enhancing the effectiveness of regenerative medicine. Studies have also found that MSCs can serve as a source of perivascular cells [34]. When cultured with endothelial cells, they increased the stability of tubular structures [35]. Cotransplanting MSCs and endothelial cells in rats rapidly generated a vascular network and stabilized blood supply [36], leading to it becoming one research focus. 

The surface features of nanomaterials can positively affect MSCs, improving their ability to differentiate, adhere, and proliferate and enhancing their bioactivity. Biomaterials must be modified to optimize them. Therefore, this study used AuNPs to modify FN and Col, creating FN-Au and Col-Au nanocomposites, and assessed their properties using pluripotent MSCs, which are widely used clinically since they can differentiate into various cell types. MSCs can be cultured and attached to the bottom of plates in vitro, forming fibroblast colony-forming units. The effects of the nanocomposites on MSC viability, morphology, differentiation, attachment, reactivity, and biological pathways were examined using various tests, including real-time polymerase chain reactions (PCR), fluorescence-activated cell sorting (FACS), immunofluorescence (IF), Alizarin Red S (ARS) staining, Oil Red-O (ORO) staining, and scanning electron microscopy (SEM). Their biocompatibility was also assessed using ROS, CD68, monocyte, platelet, filamentous (F)-actin, and cell proliferation assays. Moreover, their biological functions were explored through cell migration distance and matrix metalloproteinase (MMP) expression. Finally, their in vivo effects were evaluated through capsule formation and collagen deposition in Sprague–Dawley (SD) rats. 

This study examined the effects of the surface modification of FN and Col by AuNPs, which may enhance the biological performance and differentiation capacity of MSCs. Therefore, it explored whether the FN-Au and Col-Au nanocomposites show promise for fabricating the surface of biomedical devices.

## 2. Results

### 2.1. Characterization of the FN-Au and Col-Au Nanocomposites

The surface morphology of the four materials was analyzed using AFM (Figure 1B). In the AFM images, dark areas are deep and light areas are shallow. According to the figure, the roughness (Ra) of FN, FN-Au, Col, and Col-Au is 1.4 nm, 1.04 nm, 0.97 nm, and 1.66 nm, and distinct fibril sheets were observed in the Col-Au nanocomposite. The Ra value of the Col-Au nanocomposite (1.66 nm) was also significantly greater than that of the FN-Au nanocomposite (1.04 nm). Notably, the Col-Au nanocomposite (12.2 ppm) had the highest Ra value. The above AFM results indicate that the aggregation of the molecular chain may improve cell attachment and, in turn, promote the cell proliferation effect.

The absorption spectra of FN, Col, FN-Au, Col-Au, and pure AuNP solutions were measured using a UV-Vis spectrophotometer (Figure 1C). The change in physical structure induced in FN and Col by mixing with AuNPs reduced the molar absorption coefficient. Notably, the wavelength of AuNPs was 520 nm. After adding AuNPs to the FN and Col solutions, the pattern at 520 nm appeared, meaning that the FN-Au and Col-Au nanocomposites had been successfully fabricated and thus could be further investigated for their biocompatibility. 

The FTIR results are presented in Figure 1D, showing that pure FN exhibited the three critical patterns, indicating three specific functional groups: amide II (N-H) has a peak at 1530–1555 cm^−1^, COO^−^ (Asp and Glu) at 1565–1585 cm^−1^, and C=O at 1610–1700 cm^−1^. Moreover, the pure Col also showed three functional groups: C=O (1653 cm^−1^), N-H (1543 cm^−1^), and amide III (1240 cm^−1^). However, adding AuNPs changed the amide II band pattern, with the Col-Au nanocomposite’s N-H peak shifting from 1543 to 1534 cm^−1^. Furthermore, the FN-Au nanocomposite exhibited the same phenomenon. These results indicate that AuNPs can attract FN and Col and induce physical and structural changes. The FTIR results are consistent with the UV-Vis results, confirming the complete binding of the fabricated FN-Au and Col-Au nanocomposites. 

The SEM images of the AuNPs are shown in Figure 1E. Figure 1F shows the size distribution intensity histogram determined through DLS. Moreover, the size and diameter of AuNPs are shown in Figure 1G,H. The AuNPs had a size of 44.0 ± 4.3 and a diameter of 42.2 ± 3.8. 

### 2.2. Cell Morphology and Cytoskeleton Staining

The negative markers CD14 and CD45 were highly expressed in hematopoietic and immune cells, respectively. The positive markers CD44 and CD73 for MSCs were significantly expressed based on the flow cytometry analysis (Appendix A). After MSCs had been attached for 8, 24, and 48 h, the cytoskeleton (actin fiber) was stained with phalloidin, and cell morphology was observed using a fluorescent microscope (Figure 2A). The cells exhibited a round morphology when attached to the glass. The actin fibers extended better in the cells attached to the FN-Au and Col-Au nanocomposites, with more lamellipodia and filopodia than those on the control and the pure substrates. Furthermore, the cell attachment ability of the Col-Au nanocomposite was again superior to the FN-Au nanocomposite. As Table 1 shows, the cell size and area on the FN-Au and Col-Au nanocomposites differed significantly from the control and the pure substrates at 8, 24, and 48 h. Moreover, the results show that the FN-Au nanocomposite had a significantly greater size and area than the other groups, regardless of time, followed by the Col-Au nanocomposite, meaning that adding AuNPs effectively improved cytocompatibility. SEM was then used to further confirm cell morphology (Figure 2B). MSCs attached to the FN-Au and Col-Au nanocomposites had more actin fibers (i.e., lamellipodia and filopodia). These results suggest that migration and adhesion may improve when cells are cultured on nanocomposites.

### 2.3. Biocompatibility of FN-Au and Col-Au Nanocomposites Cultured with MSCs 

Biocompatibility is a critical issue for biomaterials after implantation into the human body. Cell viability was examined using the MTT assay, using the OD at 570 nm to compare cell viability between groups. The FN-Au and Col-Au nanocomposites both improved cell proliferation in the 24, 48, and 72 h groups (Figure 2C–E). Cell proliferation was significantly greater on the FN-Au nanocomposite than on the control, differing by 1.0-fold, 1.6-fold, and 3.6-fold in the 24, 48, and 72 h groups, respectively. It also differed significantly between the FN-Au nanocomposite and pure FN. Similarly, cell proliferation was 1.2-fold, 1.6-fold, and 3.6-fold greater on the Col-Au nanocomposite than on the control (TCPS) and differed significantly from pure Col. 

Platelets were cultured on glass coated with FN, FN-Au, Col, or Col-Au in a 24-well plate for one hour. A 2.6% glutaraldehyde/PBS solution was used to fix the platelets, which were then serially dehydrated to observe their activation using SEM (Figure 3A). The platelets showed a flattened shape on the control and the two pure substrates. In contrast, they showed a round shape (inactivated) on the two nanocomposites, with those on the Col-Au nanocomposite only showing a round morphology, indicating that it had the lowest platelet activation. The semi-quantitative results showed apparent platelet activation with the control (TCPS), while the nanocomposites exhibited less activation (Figure 3D). FN, FN-Au, Col, and Col-Au showed 0.74-fold, 0.21-fold, 0.26-fold, and 0.14-fold less activation than the control, and the latter three groups all showed a significant decrease (*p* < 0.05). Regarding the two nanocomposites, Col-Au had less activation than FN-Au, which is consistent with the SEM images.

Monocytes (10 µm) are a type of white blood cell found primarily in the peripheral blood. Once the inflammatory response is triggered, they rapidly transform into macrophages (40–50 µm). This study used the monocyte-to-macrophage transformation rate to evaluate the extent of inflammation. The more macrophages that appear, the stronger the inflammatory response. According to CD68 fluorescent staining (Figure 3B) and the semi-quantitative data (Figure 3E), the transformation rate was lower with FN (0.43-fold), FN-Au (0.19-fold), Col (0.36-fold), and Col-Au (0.18-fold) than in the control, with the Col-Au nanocomposite again having a weaker inflammatory response than the FN-Au nanocomposite. Table 2 also shows that FN (1.13-fold), FN-Au (0.98-fold), Col (1.01-fold), and Col-Au (0.94-fold) had less monocyte activation than the control, with the Col-Au nanocomposite having the weakest inflammatory response. These results suggest that adding AuNPs can decrease monocyte transformation and thus weaken the inflammatory response, showing better biocompatibility. 

MSCs were cultured on FN, Col, FN-Au, and Col-Au to observe their ROS generation. To observe their antioxidant ability, 2 × 10^3^ cells/well were cultured on each material in a six-well plate for 24 h. Flow cytometry was used to detect ROS. The results showed that ROS levels were higher in the control and two pure substrate groups than in the two nanocomposite groups (Figure 3C). Moreover, the quantitative results (Figure 3F) showed that ROS generation was significantly lower in the FN (0.84-fold), FN-Au (0.65-fold), Col (0.81-fold), and Col-Au (0.61-fold) groups than in the control group. In addition, the FN-Au and Col-Au nanocomposites differed significantly (*p* < 0.05) from the pure substrate groups (FN and Col). These results indicate that adding AuNPs significantly decreases ROS generation, confirming that the FN-Au and Col-Au nanocomposites can significantly reduce ROS generation and show outstanding biocompatibility for MSCs. 

### 2.4. MMP2 and MMP9 Zymography

Studies suggested that MMPs are closely related to cell migration since their release from cells induces them to secrete related growth factors. The zymography results (Figure 4A,B) showed that MMP2 and MMP9 expression was significantly higher on FN-Au and Col-Au. The semi-quantitative results (Figure 4D) showed that FN (1.07-fold, *p* < 0.01), FN-Au (1.23-fold, *p* < 0.01), and Col-Au (1.21-fold, *p* < 0.01) had significantly higher MMP2 expression than the control (TCPS) but not Col (1.10-fold, n.s.). Moreover, adding AuNPs induced greater MMP2 expression than the pure substrates (FN-Au, *p* < 0.01; Col-Au, *p* < 0.05). Furthermore, MMP9 showed similar trends (Figure 4E): FN (1.06-fold, *p* < 0.01), FN-Au (1.21-fold, *p* < 0.001), Col (1.10-fold, *p* < 0.05), and Col-Au (1.12-fold, *p* < 0.01). Adding AuNPs again significantly (both *p* < 0.01) improved MMP9 expression compared to the pure substrates. These zymography results for MMP2 and MMP9 indicate that the nanocomposites can induce greater MMP expression, facilitating cell migration.

### 2.5. Assessment of MSC Migration Ability

The real-time images with Calcein-AM staining were used to evaluate cell migration after culturing in various materials for 24 and 48 h (Figure 4C). The semi-quantitative migration distances (Figure 4F) indicated that they were significantly longer in the FN-Au (24 h: 2.22-fold, 48 h: 3.26-fold, *p* < 0.01), Col-Au (24 h: 2.05-fold, 48 h: 3.39-fold, *p* < 0.01), FN (24 h: 2.19-fold, 48 h: 2.86-fold, *p* < 0.01), and Col (24 h: 1.94-fold, 48 h: 2.90-fold, *p* < 0.01) groups than in the control group at both 24 and 48 h. Additionally, they were longer in the FN-Au and Col-Au groups than in the FN and Col groups at 48 h, meaning that adding AuNPs could increase cell migration, making it suitable for tissue repair.

### 2.6. Integrin Expression: α2, α4, and αVβ3

Integrins α2, α4, and αVβ3 are receptors for cells to attach to the ECM and are related to angiogenesis and cell signal communication. These processes result in cell differentiation and proliferation, which are responsible for vessel repair and regeneration. Cells were cultured on TCPS, FN, FN-Au, Col, and Col-Au to observe their integrin expression after 48 h. The results show that the expression of integrins α2, α4, and αVβ3 was significantly higher in MSCs cultured on the FN-Au and Col-Au nanocomposites than on the control and pure substrates (Figure 5). Integrin α2 expression differed between the FN (1.01-fold), FN-Au (1.39-fold), Col (0.98-fold), and Col-Au (1.74-fold) groups and the control group, differing significantly (*p* < 0.05) between the nanocomposites and the pure substrates. In addition, integrin α4 expression was significantly higher in the FN (1.04-fold), FN-Au (1.28-fold), Col (1.06-fold), and Col-Au (1.66-fold) groups than in the control (TCPS) group, differing more significantly for the two nanocomposite groups than for the two pure substrates (FN-Au, *p* < 0.05; Col-Au, *p* < 0.001). Moreover, integrin αVβ3 expression was significantly higher in all experimental groups than in the control (TCPS) group. Notably, the difference was again more significant in the FN-Au and Col-Au groups than in the pure substrate groups (FN-Au, *p* < 0.01; Col-Au, *p* < 0.05). These results demonstrate that the FN-Au and Col-Au nanocomposites can improve cell migration and attachment. 

### 2.7. Assessments of the Multi-Differentiation Capacity of MSCs

Real-time PCR was used to observe the expression of various genes in MSCs cultured on the different materials for 3, 5, and 7 days (Figure 6). The MSC culture methods and protocols were the same as those used above. Each gene has a specific role: glial fibrillary acidic protein (*GFAP*), neuroepithelial stem cell protein (nestin [*NES*]), and tubulin were used to assess neural differentiation; peroxisome proliferator-activated receptor (*PPAR*) was used to assess adipogenesis; Runt-related transcription factor 2 (*RUNX2*) was used to assess osteogenesis; and platelet and endothelial cell adhesion molecule 1 (*PECAM1*/*CD31*) and von Willebrand factor (*VWF*) were used to assess angiogenesis. The PCR results for all these genes showed the same pattern (FN-Au and Col-Au > FN and Col > TCPS (control), indicating that the nanocomposites have a significantly better capacity to induce MSC differentiation into neuron, adipose, bone, and endothelial cells. 

IF staining was performed to confirm the protein expression of the three neural proteins (GFAP, NES, and tubulin) and the two endothelial proteins (CD31 and VWF; Figure 7A–E). Similar to the real-time PCR results, the semi-quantitative data (Figure 7F–J) showed that their protein expression was higher in the FN-Au and Col-Au groups than in the control and pure substrate groups. In addition, their expression was significantly higher in the FN-Au and Col-Au groups than in the control (TCPS) group after seven days (*p* < 0.05), except for VWF in the FN-Au group (n.s.). These results confirmed the real-time PCR results and indicated that the FN-Au and Col-Au nanocomposites can promote MSC differentiation and proliferation. The fluorescent images and semi-quantitative results for three and five days are shown in Appendix A.

After examining the expression of seven genes (three neural, one adipose, one bone, and two endothelial) using real-time PCR, we further confirmed the protein expression of the neural and endothelial proteins using IF staining. Next, we examined calcium deposition and neutral lipids using ARS and ORO staining after 3, 5, 7, and 10 days. Note that the cell culturing protocols were the same as those used above. 

Regarding osteogenic induction, the mineral deposition measured based on ARS staining was significant in the FN-Au and Col-Au groups (Figure 8A). The semi-quantitative data showed that the FN-Au and Col-Au groups had the highest calcium deposition after seven days and were significantly higher than the control and the pure substrate groups (Figure 8C). The FN-Au group differed significantly (*p* < 0.01) from the FN group and the Col-Au group (*p* < 0.05). In addition, calcium deposition was significantly higher in the FN-Au and Col-Au groups than in the control, FN, and Col groups at 3, 5, 7, and 10 days, indicating that nanocomposites can indeed improve osteoblastic differentiation (Appendix A). 

Regarding adipocyte differentiation, intracellular neutral lipid contents measured based on ORO staining were significant in the FN-Au and Col-Au groups on day 7 (Figure 8B,D). While the FN and Col groups did not differ significantly from the control (TCPS) group on day 5, the FN-Au and Col-Au groups had significantly greater (*p* < 0.01) neutral lipid amounts on day 10. In addition, the two nanocomposite groups differed significantly from the pure substrate groups. Moreover, the FN-Au group differed significantly (*p* < 0.01) from the FN and Col-Au groups (*p* < 0.001). The results for days 3, 5, and 10 are shown in Appendix A. The semi-quantitative results indicate that neutral lipid amounts increased daily and to a greater extent in the FN-Au and Col-Au groups, proving that the nanocomposites can effectively promote adipogenesis.

### 2.8. Cell Cycle Analysis and the Expression of Apoptotic-Related Proteins in MSCs 

The impact of FN, FN-Au, Col, and Col-Au on the cell cycle in MSCs was investigated using flow cytometry (Figure 9A). As shown in Figure 9B, the population of MSCs in the sub-G1 phase was significantly lower (*p* < 0.05) in the FN-Au, Col, and Col-Au groups, with the Col-Au group even lower than the pure Col group (n.s.). In contrast, the population of MSCs in the S phase was significantly higher in the FN-Au (*p* < 0.001) and Col-Au (*p* < 0.01) groups and differed significantly from the pure substrate groups (FN: *p* < 0.05; Col: *p* < 0.01). 

Annexin V/PI double staining also demonstrated that the FN-Au and Col-Au nanocomposites could significantly reduce the ratio of apoptotic MSCs (*p* < 0.001) and also differed significantly from the pure substrate groups (*p* < 0.05). These results indicate that the FN-Au and Col-Au nanocomposites could prevent MSCs from undergoing apoptosis (Figure 9C,D).

WB was used to examine the expression of apoptotic-related proteins, such as B-cell leukemia/lymphoma 2 (BCL2), BCL2-associated X apoptosis regulator (BAX), activated caspase 3 (CASP3), p21, and cyclin D1 (CCND1; Figure 10A). BAX, CASP3, and p21 are so-called “apoptosis-induced proteins”, while BCL2 and CCND1 are “anti-apoptotic proteins”. Each protein was semi-quantified based on its expression intensity. The expression levels of the pro-apoptotic proteins BAX, activated CASP3, and p21 were significantly lower in the FN-Au and Col-Au groups (*p* < 0.001, except for BAX and CASP3 in the Col-Au group [*p* < 0.01]; Figure 10B–D). Moreover, the FN-Au and Col-Au groups both differed significantly from the pure substrate groups. Regarding the anti-apoptotic proteins, BCL2 and CCND1, all groups, including FN, FN-Au, Col, and Col-Au, showed significantly lower BCL2 expression than the control group (*p* < 0.001), and the nanocomposites again had significantly lower expression than the pure substrate groups (Figure 10E). CCND1 expression did not differ significantly in FN, FN-Au, and Col groups but was significantly higher in the Col-Au group (*p* < 0.05) compared to the control group (Figure 10F). While the Col-Au nanocomposite improved CCND1 expression, the BCL2 results were unexpected and require further investigation. Overall, while the FN-Au and Col-Au nanocomposites both decreased pro-apoptotic protein expression, only the Col-Au nanocomposite increased CCND1 expression. These results indicate that the nanocomposites effectively inhibited apoptotic processes.

### 2.9. Assessment of In Vivo Biocompatibility

The inflammatory response was investigated after the various materials had been subcutaneously implanted in SD rats for one month to determine the clinical potential of the nanocomposites. Tissue staining was used to examine capsule formation (Figure 11A) and collagen deposition (Figure 11B). The H&E staining results are shown in these figures, with the small yellow double arrows indicating capsule thickness. The semi-quantitative results (Figure 11C) indicated that capsules were significantly thinner in the FN-Au and Col-Au groups than in the control group (*p* < 0.001) and pure substrate groups (*p* < 0.01). Regarding collagen deposition (Figure 11D), which was visualized via Masson’s trichrome staining, the quantitative results showed significantly lower (*p* < 0.001) collagen deposition in the FN-Au and Col-Au groups than in the control and pure substrate groups (*p* < 0.01). 

CD86 and CD163 were selected as M1 and M2 macrophage polarization markers to evaluate the inflammatory response after subcutaneous implantation in SD rats. Their expression levels were semi-quantified based on the fluorescence intensity (Figure 12A,B). The results demonstrated that CD86 expression (Figure 12D) was lowest in the Col-Au group (0.63-fold, *p* < 0.001), followed by the FN-Au group (0.69-fold, *p* < 0.001), and was significantly lower in the Col-Au and FN-Au groups than in the FN (*p* < 0.01) and Col groups (*p* < 0.001). CD163 expression (Figure 12E) was highest in the Col-Au group (1.49-fold, *p* < 0.05), followed by the FN-Au group (1.40-fold, *p* < 0.01), and was significantly higher in the Col-Au and FN-Au groups than in the FN and Col groups (*p* < 0.01). Moreover, the endothelialization marker CD31 was also examined (Figure 12C), and its expression was higher in the experimental groups than in the control group. Notably, CD31 expression was significantly higher in the Col-Au (1.37-fold, *p* < 0.05) and FN-Au (1.31-fold, *p* < 0.01) groups than in the control (TCPS) and pure substrate groups (Figure 12F). The scheme for the experimental abstract for this article is shown in Figure 13. 

## 3. Discussion

FN is a complex high-molecular-weight (470–500 kDa) glycoprotein localized to the ECM [37], and collagen is a critical protein in human bodies with outstanding biocompatibility and biodegradability [38]. Since these molecules can stimulate angiogenesis in vitro and in vivo, they are widely used to study the ECM [39]. Research also indicates that materials made from gold nanocomposites can enhance cell attachment, improving cell proliferation [40]. To create a more biocompatible microenvironment for cell growth and examine the biological effect of culturing MSCs on these materials, we coated glass with four nanomaterials (FN, FN-Au, Col, and Col-Au) and examined their effects on MSCs.

The AFM results showed that adding AuNPs significantly decreased the Ra value of FN from 1.4 nm to 1.04 nm due to the flattened molecular chain, which reduces the surface area. In contrast, the results for collagen differ, with the Ra value significantly higher for the Col-Au nanocomposite (1.66 nm) than Col (0.97 nm), indicating that the aggregation of molecule chains increases the surface area. Previous studies also showed that adding AuNPs altered the structure of the molecule chain and, in turn, the cell attachment ability [12]. According to the UV-Vis results, the FN-Au and Col-Au nanocomposites had an absorption peak at 520 nm, proving that the AuNPs were bound to the biomolecules [41]. FTIR was also used to detect the functional groups of each material, showing that FN had three specific peaks at 1530–1555 cm^−1^ (NH), 1565–1585 cm^−1^ (COO^−^), and 1610–1700 cm^−1^ (C=O), while the FN-Au nanocomposite showed a peak at 1543 cm^−1^ (NH) [14]. Moreover, Col also had three specific peaks at 1653 cm^−1^ (C=O), 1543 cm^−1^ (NH), and 1240 cm^−1^ (amide III), while the Col-Au nanocomposite had a peak at 1534 cm^−1^. These results indicate that the biomolecule-AuNP composites were successfully fabricated [12]. 

Previous studies have confirmed that AuNP composites have outstanding biocompatibility [42]. In this study, the four materials were coated on glass, and then, MSCs were cultured to observe cell proliferation. The results demonstrated that MSCs proliferated better on the FN-Au and Col-Au nanocomposites. The AuNPs can also affect hydrogen bonds, softness, and ROS removal [43,44]. Our previous studies showed that nanocomposites can prevent oxidation and inflammation and slow degradation. CD68 was used as a macrophage marker to examine the anti-inflammatory abilities of the nanocomposites. The results indicated that the macrophage conversion rate was significantly lower in the FN-Au and Col-Au groups. In addition, many studies have shown that ROS can lead to DNA damage, including replication error, single or double helix break, gene instability, and oncogenesis [45]. Our results showed that the FN-Au and Col-Au groups had the lowest ROS levels, suggesting that AuNPs (12.2 ppm) could enhance biostability and biocompatibility.

Our results showed a round morphology when the MSCs were attached to the glass, indicating poor growth. However, when the MSCs were attached to the FN-Au and Col-Au nanocomposites, the actin fibers had better ductility, with more lamellipodia and filopodia than in the control and pure substrate groups [12,46]. Furthermore, the morphology of attached cells was observed using SEM, which supported the above results. Previous studies have also reported that cells may attach better to gold nanocomposites. Our results showed that the nanocomposites could increase the adhesion area and are thus suitable for cell migration and attachment [12].

Integrins α2, α4, and αVβ3 are receptors that allow cells to attach to the ECM or even each other. These integrins are associated with angiogenesis and indirectly influence cell signaling, leading to differentiation, proliferation, and, in turn, angiogenesis [47]. Our results showed that integrin α2, α4, and αVβ3 expression was significantly higher in MSCs cultured on the FN-Au and Col-Au nanocomposites than on the other substrates without AuNPs, proving that they can improve cell migration ability and attachment [48]. To further observe the cell migration effect, a migration assay and fluorescent microscope were used to evaluate the biological functions of the nanocomposites for 24, 48, and 72 h [12]. Our results confirmed that FN, FN-Au, Col, and Col-Au improved cell migration, with the improvements greater with the FN-Au and Col-Au nanocomposites, consistent with the above results [13]. Gelatin zymography was also used to detect MMP expression, including MMP2 and MMP9, which are believed to improve migration. Our results showed that adding AuNPs again increased the MMP2 and MMP9 expression [49,50]. All of these results show that the FN-Au and Col-Au nanocomposites can enhance cell migration. Moreover, their biocompatibility was evaluated through WB, flow cytometry, and Annexin V/PI double staining assays. Our results showed that the FN-Au and Col-Au nanocomposites significantly reduced pro-apoptotic protein levels, significantly decreasing MSC apoptosis (*p* < 0.001), demonstrating the safety and stability of these nanocomposites. 

This study used MSCs due to their multiple-differentiation and outstanding proliferation abilities [51]. Previous research indicates that MSCs can be isolated from various tissues and induced into various cell types [52]. In this study, we aimed to develop two types of nanocomposites, FN-Au and Col-Au, and examine whether MSCs can grow well on them and differentiate into neuron, osteoblast, adipocyte, and endothelial cells (nerve, bone, fat, and vascular, respectively). Real-time PCR, ARS, and ORO were used to assess genes and proteins related to these differentiation events. Seven genes were investigated in the real-time PCR experiment. GFAP, NES, and tubulin are neurogenesis markers. GFAP is found in astrocytes that usually exist in the central nervous system in mammals [53]. Tubulin is regarded as an early marker of neuron differentiation [54]. NES is a type VI intermediate filament protein in neurons and is also considered a marker of neural stem cells [55]. These genes (*GFAP*, *NES*, and tubulin) all showed significant expression in MSCs cultured on the FN-Au and Col-Au nanocomposites, of which tubulin showed the highest expression. *CD31* and *VWF* were used as endothelial markers, *PPAR* as an adipocyte marker, and *RUNX2* as an osteoblast marker. All of these results showed that adding AuNPs significantly affected MSC differentiation. These results were then further investigated via IF, ARS, and ORO staining to confirm the translation of these proteins. The three neural and two endothelial markers were assessed through IF staining, while ARS was used to assess calcium deposition [56], and ORO was used to assess adipose tissue [57]. Interestingly, *GFAP*, tubulin, and *NES* expression was greater with the Col-Au nanocomposite, while CD31 and VWF expression was greater with the FN-Au nanocomposite [58]. These results indicated that the FN-Au and Col-Au nanocomposites could induce MSC differentiation into neural [59], bone [60], adipocyte [61], and endothelial [62] cells, making them promising materials for future clinical use.

This study intended to provide a functionalized surface coating comprised of AuNPs and FN or Col to improve the biocompatibility and favorable cellular response of MSCs on the nanocomposites. The surface morphology of FN and Col was markedly changed by adding a small amount of AuNPs. The FN-Au and Col-Au nanocomposites exhibited better biocompatibility and biological performance than the pure FN and Col, including promoting cell proliferation, reducing platelet and monocyte activation, and reducing ROS generation. In addition, the FN-Au and Col-Au nanocomposites induced better MSC differentiation, making them promising materials for tissue regeneration.

Biocompatibility is a critical concern for biomaterials implanted into the human body. This study demonstrated that the FN-Au and Col-Au nanocomposites elicited a weaker foreign body reaction than pure FN and Col. In addition, the FN-Au and Col-Au nanocomposites significantly attenuated platelet activation, potentially reducing thrombus formation, and showed good blood compatibility. Moreover, their better attenuation of ROS generation may decrease inflammation after implantation. This study assessed the nanocomposites in vitro and in vivo using SD rats as animal models. Capsule formation and collagen deposition were both significantly lower with the nanocomposites. The CD86 and CD163 macrophage polarization markers revealed that M2 polarization predominated in the nanocomposite groups, indicating that tissue repair occurs before inflammation and phagocytosis. Furthermore, the CD31 results were consistent with the endothelial differentiation markers. Therefore, this study successfully fabricated two nanomaterials, FN-Au (12.2 ppm) and Col-Au (12.2 ppm), with better stability, biocompatibility, and differentiation capacity in vitro and in vivo. However, these materials require further investigation and evaluation. We hope to conduct a clinical trial and develop promising nanomaterials based on them for clinical use.

## 4. Materials and Methods

### 4.1. Materials Characterization 

#### 4.1.1. Preparation of FN-Au and Col-Au Nanocomposites

This study used AuNPs (Gold Nanotech, Taipei, Taiwan) with a 3–5 nm diameter. The AuNP solutions were passed through a 0.22 µm filter to create an aseptic solution. A 40 µg/mL FN (Millipore, NJ, USA) solution was prepared by dissolving 1 mg of FN in 24 mL of PBS buffer. The collagen (Col; Millipore, NJ, USA) solution was prepared by dissolving 4.1 mg of Col in 24 mL of deionized water. We then added 244 µL of AuNPs to 9756 µL of the FN or Col solution to prepare a 12.2 ppm FN-Au or Col-Au solution. After thorough mixing, we coated the nanocomposite solutions onto a 96-well, 24-well, 6-well, and 10 cm dish for 30 min. Then, we removed the supernatant to obtain the nanocomposite thin-film materials for further investigation. A brief scheme of the materials preparation is shown in Figure 1A.

#### 4.1.2. Atomic Force Microscopy (AFM)

After drying at room temperature for 12 h, the nanocomposites coated onto the round glass were analyzed using AFM with a 100 µm piezoelectric scanner (JSM-6700F; JEOL, Tokyo, Japan) to observe MSC morphology. The images were taken in tapping mode, using a triangular cantilever with a force constant of 21–78 N/m that supported an integrated pyramidal tip of Si3N4. Finally, the average roughness of the material surface was analyzed using Image-Pro Plus 4.5 software.

#### 4.1.3. Fourier-Transform Infrared Spectroscopy Analysis (FTIR)

The coating was also analyzed using FTIR (IRPrestige-21; Shimadzu, Kyoto, Japan). After mixing each group with potassium bromide (KBr) and pressurizing it in an ingot, each sample was scanned eight times in the spectral region of 400–4000 cm^−1^ with a resolution of 2 cm^−1^. These results were then averaged to produce the spectrum for the four groups to examine the differences after surface modifications.

#### 4.1.4. UV-Visible Spectrophotometry (UV-Vis)

Different ultraviolet and visible light energies can be visualized in a UV-Vis spectrum (Thermo Fisher Scientific, Waltham, MA, USA). After cleaning the instrument, deionized water was used to determine the background, and then, each sample was measured in the wavelength from 190 to 1100 nm. Note that the AuNPs have a characteristic peak at 520 nm. Finally, the experimental data were quantified using OriginLab 8 software (OriginLab Corporation, Northampton, MA, USA). 

#### 4.1.5. Dynamic Light Scattering Assay (DLS)

DLS was used to measure the size and diameter of the AuNP particles with a Malvern Zetasizer Nano ZS device (Malvern Panalytical Ltd., Malvern, UK) using a 532 nm light source at a 90° fixed scatter angle. Briefly, 1 mL of the AuNP solution was added to an optical path cuvette and then analyzed using the Zetasizer Nano ZS with a 633 nm Helium-Neon laser to determine the intensity distribution at 25 °C. Finally, these data were analyzed using Malvern Zetasizer software (version 7.1, Malvern Panalytical Ltd.).

### 4.2. Biocompatibility Assessment

#### 4.2.1. Culturing of Human UCMSCs

Human umbilical cord (Wharton’s jelly)-derived mesenchymal stem cells (MSCs) were kindly provided by Prof. Woei-Cherng Shyu (China Medical University, Taiwan). The MSCs used in this study were derived from the human umbilical cord and had been passaged 8–20 times. The cells were cultured in a 10 cm^2^ dish containing high-glucose Dulbecco’s modified Eagle’s medium (H-DMEM; Invitrogen, Waltham, MA, USA) supplemented with 10% FBS (Invitrogen), 100 U/mL penicillin/streptomycin (Invitrogen), and 1% sodium pyruvate (Invitrogen). After the cells had proliferated to cover about 80% of the dish, which took 2–3 days, they were placed in an incubator at 37 °C and 5% CO_2_ (Thermo Forma 370; Waltham, MA, USA) for further passaging. A 15 mm coverslip was first coated with the test materials, seeded with MSCs at a concentration of 2 × 10^4^ cells per 24-well plate, and then incubated in the conditioned medium for subsequent studies. Cells at a density of 2 × 10^5^ cells per well were seeded into the cell culture plates coated with the different nanomaterials and then incubated in a growth medium for eight hours. Next, the supernatant was removed and replaced with a 10% bovine serum albumin-free culture medium for 24 h. Then, the differentiation of MSCs on these plates was observed. The specific surface markers of MSCs were characterized via flow cytometry. MSCs were incubated with antibodies against these markers conjugated with fluorescein isothiocyanate (FITC) or phycoerythrin (PE): CD14-FITC, CD45-FITC, CD44-PE, and CD73-PE (BD Pharmingen, San Diego, CA, USA). The isotype controls were PE-conjugated IgG1 and FITC-conjugated IgG1 (BD Pharmingen). The MSC phenotypes were analyzed using FACS software (LSR II; Becton Dickinson, MA, USA). 

#### 4.2.2. MTT Assay

The effects of the materials (TCPS, FN, FN-Au, Col, and Col-Au) on cell growth were determined using the MTT assay. Cells (200 μL/well at a density of 2 × 10^4^ cells/mL) were cultivated with medium containing the test samples in a 96-well culture plate at 37 °C with 5% CO_2_. After 24, 48, and 72 h of incubation, the medium was removed, 100 μL of MTT solution (0.5 mg/mL) was added to each well, and the plate was placed in an incubation room at 37 °C for 2–4 h. Next, the MTT solution was removed, and 100 μL of DMSO was added for 15–30 min. Finally, the OD value of each well at 570 nm was determined using an ELISA reader to measure cell growth. The cell growth rate (%) was calculated using the following formula: (%) = [(OD1 − OD0)/(OD2 − OD0)] × 100, where OD0, OD1, and OD2 represent the mean OD of the blank, experimental, and control groups, respectively. All experiments were performed in triplicate. 

#### 4.2.3. Platelet Activation Test

Glass coverslips were coated with the materials and placed in a 24-well culture plate containing 0.5 mL of platelet-rich plasma (2 × 10^6^ platelets/mL) per well. After a one-hour incubation, the coverslips were removed, and the number of adherent platelets was counted using a cell counter (Assistant, Königswinter, Germany). The coverslips were incubated at 37 °C under 5% CO_2_ for one hour and then fixed with 2.5% glutaraldehyde (Sigma Aldrich, Saint Louis, MO, USA) for at least eight hours. Next, the samples were dehydrated in serial ethanol concentrations (30–100%) and then dried to the critical point. Finally, the samples were observed under an SEM (JEM-5200; JEOL, Pleasanton, CA, USA) to identify platelet activation based on their shape changes. The degree of platelet activation used a score ranging from 0 (round, inactive) to 1 (fully spread, totally activated). The average platelet activation on various materials was calculated based on the references.

#### 4.2.4. Monocyte Activation Test

Human monocytes were obtained from healthy adult volunteers at Taichung Veteran General Hospital. This protocol was approved by the Institutional Review Board (approval number CE12164). Monocytes were isolated by mixing 20 mL of whole blood at 1:1 with PBS buffer and then centrifuging the solution with 3 mL of Ficoll for 20 min (X-22R; Beckman Coulter, MA, USA). Then, the blood plasma layer was removed, and the buffy coat was washed twice with PBS buffer. The living cells were observed and counted using trypan blue. Next, we added cells at 10^5^/mL in standard medium (containing 10% FBS and 1% [*v*/*v*] antibiotics [10,000 U/mL penicillin G and 10 mg/mL streptomycin]) to a 24-well plate with TCPS, FN, Col, FN-Au, and Col-Au, cultured them in an incubator at 37 °C with 5% CO_2_ for 96 h, and then separated them from the wells using trypsin. Finally, we used a microscope to calculate the monocyte conversion yield using Equation (1).
(1)$$\mathrm{Conversion}\;\mathrm{yield}=\frac{Macrophage}{\left(Monocyte+Macrophage\right)}\;\times \;100\%$$

#### 4.2.5. Cell Morphology and Adhesion Ability

SEM was used to analyze the morphology and attachment ability of 1 × 10^4^ MSCs on different materials. The cells were cultured at 37 °C in a 5% CO_2_ incubator for 48 h and then fixed with a 2.5% glutaraldehyde solution for at least eight hours. After being washed twice with PBS, they were dehydrated using a series of ethanol concentrations (30–100%). Finally, the cells were critical-point-dried, and SEM was used to observe their morphology on each material.

#### 4.2.6. Measurement of Intracellular ROS

2,7-Dichlorofluorescein diacetate (DCFDA) (Sigma Aldrich, Saint Louis, MO, USA) was used to measure the ROS production inside cells. MSCs (2 × 10^5^/well) were cultured in six-well plates coated with FN, FN-Au, Col, or Col-Au for 48 h. After washing the cells twice with PBS, 500 mL of PBS containing 20 mM DCFDA was added and incubated at 37 °C for 60 min. Then, a FACSCalibur™ Flow Cytometer (Becton Dickinson, Franklin Lakes, NJ, USA) was used to measure the emission at 530 nm after excitation at 480 nm at 30 min intervals for four hours. Note that higher fluorescence intensity indicated greater intracellular ROS production. Finally, fluorescein-positive cells were evaluated using FCS software (Becton Dickinson, Franklin Lakes, NJ, USA).

#### 4.2.7. Fluorescent Staining of the Cytoskeleton

Twenty-four-well plates coated with TCPS, FN, FN-Au, Col, or Col-Au were seeded with 1 × 10^4^ MSCs and incubated at 37 °C in a 5% CO_2_ incubator for 8, 24, or 48 h. Next, the cells were washed thrice with PBS and fixed with 4% paraformaldehyde for 10 min. Then, they were incubated with 0.5% (*v*/*v*) Triton X-100 (Sigma Aldrich, Saint Louis, MO, USA) in PBS at room temperature for 10 min and then with phalloidin (Sigma Aldrich, Saint Louis, MO, USA) diluted 1:250 with PBS at room temperature in the dark for 60 min. Next, they were stained with 4′-6-diamidino-2-phenylindole (DAPI; Invitrogen, Waltham, MA, USA) for 10–20 min under the same conditions. Finally, the glass slide was covered with a 15 mm glass coverslip, sealed with mounting gel (Invitrogen, Waltham, MA, USA), and covered with tin foil to block the light, and the cells’ cytoskeleton morphology was observed and recorded under a fluorescent microscope (Axio Imager A1; Zeiss, Danvers, MA, USA).

### 4.3. Biological Function Evaluation

#### 4.3.1. Gelatin Zymography Analysis

First, 2 × 10^5^ cells were seeded in a 10 cm^2^ culture dish and cultured at 37 °C in a 5% CO_2_ incubator for 48 h. Next, the medium was collected to detect the metalloproteinases. A 10% sodium dodecyl sulfate-polyacrylamide electrophoresis (SDS-PAGE) gel was prepared using ddH_2_O, Tris buffer, acrylamide/bis-acrylamide (29:1; SERVA, Heidelberg, Germany), ammonium persulfate (APS; J.T. Baker, Stockbridge, GA, USA), 2% gelatin (Invitrogen, Waltham, MA, USA), and tetramethylethylenediamine (TEMED; J.T. Baker, Stockbridge, GA, USA). After the samples were loaded, SDS-PAGE was conducted with the Mini PROTEAN System (J.T. Baker, Stockbridge, GA, USA) at 80–120 V for about 90–120 min. Next, the gel was washed three times with a washing buffer (40 mM Tris-HCl [pH 8.5], 0.2 NaCl, 10 mM CaCl_2_, and 2.5% Triton X-100) at room temperature and then incubated with the reaction buffer (40 mM Tris-HCl [pH 8.5], 0.2 NaCl, and 10 mM CaCl_2_; Invitrogen, Waltham, MA, USA) and 0.01% NaN_3_ (Invitrogen, Waltham, MA, USA) at 37 °C for 12 h. Then, the gel was stained with a Coomassie blue solution (0.2% Coomassie blue R-250, 50% methanol, and 10% acetic acid; Sigma-Aldrich, Saint Louis, MO, USA) for 30 min and destained with 10% acetic acid and 20% methanol. Finally, the gel was scanned, and protein levels were quantified using Gel-Pro analyzer 4.0 (Media Cybernetics, GA, USA).

#### 4.3.2. Real-Time PCR 

MSCs (2 × 10^5^/well) were cultured in six-well culture plates coated with the different materials for 3, 5, or 7 days under standard conditions (37 °C and 5% CO_2_). After incubation, the medium was removed, and total RNA was extracted from the MSCs using TRIzol reagent (Invitrogen, Waltham, MA, USA) according to the manufacturer’s instructions. Briefly, 1 mL of TRIzol solution was added to each well and left for five minutes. Next, 200 μL of chloroform (Sigma-Aldrich, Saint Louis, MO, USA) was added to each well and left for 15 s. Then, the culture plates were left to stand at room temperature for three minutes. Next, the RNA was pelleted via centrifugation at 12,000 rpm and 4 °C for 15 min. Then, the supernatant was discarded, and 500 μL of isopropanol was added and incubated at 4 °C for 10 min. Next, the plates were centrifuged at 12,000 rpm and 4 °C for 15 min, the supernatant was discarded, and the RNA pellet was washed twice with 1 mL of 75% alcohol and then left to dry. Next, 20 μL of DEPC-treated H_2_O was added to each well, and its absorbance at 260 nm was measured using an ELISA plate reader (SpectraMax M2; Molecular Devices, San Jose, CA, USA). Then, cDNA was synthesized using the RevertAidTM First Strand cDNA Synthesis Kit (Fermentas, Burlington, ON, Canada) according to the manufacturer’s instructions. Finally, the mRNA levels of the studied genes in MSCs after various treatments were quantified using a Step OneTM Plus Real-Time PCR System.

#### 4.3.3. Western Blotting (WB)

First, cells (2 × 10^5^/well) were seeded into a 10 cm^2^ cell culture plate and cultured at 37 °C in a 5% CO_2_ incubator for 48 h to allow them to attach. Next, the cells in each group (TCPS, FN, FN-Au, Col, and Col-Au) were transferred into the centrifuge tubes containing 0.05% trypsin-EDTA (Invitrogen, MA, USA), washed twice with PBS, and then incubated with a cell lysis buffer at 4 °C for 60 min. Finally, the tubes were centrifuged at 13,000 rpm and 4 °C for 20 min, and the supernatant containing the total cellular proteins was collected. Next, the protein concentrations of the samples were determined using bicinchoninic acid (BCA; Bio-Rad, Hercules, CA, USA), which reacts with protein to form a dark purple compound that absorbs light at 595 nm. First, a 0.32 mg/mL bovine serum albumin (BSA) solution was prepared, and BCA was diluted 4:1 with ddH_2_O. Next, the appropriate amount of BSA was mixed with BCA to prepare 3.2, 1.6, 0.8, 0.4, and 0 µg/mL solutions, which were measured at 595 nm using an ELISA plate reader to construct a standard curve. Note that the ratios of the running gel varied based on the molecular weight. Next, the separated proteins were transferred to a PVDF membrane using a wet Mini Trans-Blot Cell (Bio-Rad, Hercules, CA, USA) with a transfer buffer (Tris-base, glycine, and 100% methanol [J.T. Baker, Stockbridge, GA, USA] in ddH_2_O) at 4 °C for one hour and then immersed in TBST (Tris-HCl and Tween-20 [J.T. Baker, Stockbridge, GA, USA] with NaCl in ddH_2_O) containing 5% skimmed milk powder at room temperature with shaking for one hour for blocking. Then, the quantified total proteins were mixed with a loading buffer and separated via SDS-PAGE using the Mini PROTEAN System (Bio-Rad, Hercules, CA, USA) at 80–120 V for about 90–120 min. Next, the separated proteins were transferred to a PVDF membrane using a wet Mini Trans-Blot Cell (Bio-Rad, Hercules, CA, USA) with a transfer buffer (Tris-base, glycine, and 100% methanol [J.T. Baker, Stockbridge, GA, USA] in ddH_2_O) at 4 °C for one hour and then immersed in TBST (Tris-HCl and Tween-20 [J.T. Baker, Stockbridge, GA, USA] with NaCl in ddH_2_O) containing 5% skimmed milk powder at room temperature with shaking for one hour for blocking. Then, the membrane was incubated with the primary antibodies against BAX, BCL2, caspase 3, cyclin D1, and p21 (1:1000; Santa Cruz Biotechnology, Santa Cruz, CA. USA) and β-actin (1:5000) in TBST at 4 °C overnight. Next, the membrane was washed four times with TBST (10 min each) before being incubated with the secondary antibodies (1:5000) for one hour. Then, the membrane was washed four times with TBST (10 min each). Finally, the membrane was incubated with the enhanced chemiluminescence (ECL) reagent (PerkinElmer, MA, USA), and the protein bands were observed in a dark room and analyzed using Gel-Pro Analyzer 4.0 software.

#### 4.3.4. Cell Migration Assay

The MSCs were seeded (1 × 10^4^/well) into a 96-well plate with stoppers inserted from the Cell Migration Assay Kit (Platypus Technologies, Fitchburg, WI, USA) and cultured at 37 °C in a 5% CO_2_ incubator. After cell attachment, the stoppers were removed, except for one well (pre-migration). Next, the seeded plates were incubated at 37 °C to measure pre-migration (*t* = 0 h) and post-migration (*t* = 24 and 48 h). Then, Calcein AM (1:2000; Invitrogen, Waltham, MA, USA) was added to the wells representing the two post-migration time points and incubated for 30 min. Finally, the cell migration distance was observed under a fluorescent microscope and semi-quantitatively analyzed using ImageJ 5.0 software.

#### 4.3.5. IF Assay

MSCs were seeded onto glass coverslips (1 × 10^4^) coated with the FN, FN-Au, Col, and Col-Au materials and cultured at 37 °C in a 5% CO_2_ incubator. After 3, 5, and 7 days, the medium was removed, and the attached cells were washed thrice with PBS, fixed with 4% PFA (Sigma-Aldrich, Saint Louis, MO USA) for 15 min, and then washed with PBS to remove the remaining PFA. Next, the cells’ plasma membranes were lysed using 0.1% Triton X-100 (J.T. Baker, Stockbridge, GA, USA) and washed thrice with PBS, and 5% FBS was added for blocking. Then, the cells were incubated with primary antibodies against CD31, CD68, GFAP, nestin, tubulin, and VWF (1:250; Santa Cruz Biotechnology, Santa Cruz, CA, USA) at 4 °C for at least eight hours. Next, they were washed twice with PBS and then incubated with fluorescence isothiocyanate (FITC)-conjugated goat anti-mouse and anti-rabbit secondary antibodies (Jackson ImmunoResearch, West Grove, PA, USA) at room temperature for one hour. Then, they were incubated with 4,6-diamidion-2-phenylindole (DAPI; Invitrogen, Waltham, MA, USA) for 10 min in the dark. Finally, the coverslips were prepared as described in Section 4.2.7 and observed under a fluorescent microscope.

#### 4.3.6. Cell Cycle and Apoptosis Examination

First, propidium iodide (PI; Sigma-Aldrich, Saint Louis, MO, USA) was used to stain the cell nuclei. Then, the cell cycle progression of the cells was investigated using a flow cytometer (LSR II; BD, CA, USA). Apoptotic cells were investigated using annexin V (ANXA5) and PI double staining (Invitrogen, Waltham, MA, USA) according to the manufacturer’s instructions. At the early stage of apoptosis, phosphatidylserine (PS) exposed on the outside of the plasma membrane can be bound by annexin V. At the late stage of apoptosis, PI can bind to the chromosomes of permeabilized cells. This staining was observed under a fluorescent microscope and semi-quantified using ImageJ 5.0 software.

### 4.4. Cell Differentiation Examination

#### 4.4.1. Flow Cytometric and FACS Analysis

The protein levels of various integrins were examined by seeding 2 × 10^4^ cells in six-well plates coated with FN, FN-Au, Col, or Col-Au and incubating them at 37 °C in a 5% CO_2_ incubator for 48 h. Next, 0.05% trypsin-EDTA was used to detach the cells, which were transferred into centrifuge tubes, washed thrice with PBS, and then incubated with primary antibodies against the integrin α2 (ITGA2) and α4 (ITGA4) subunits and αVβ3 complex (Santa Cruz Biotechnology, Santa Cruz, CA, USA) at room temperature for one hour. Finally, the cells were incubated with FITC-conjugated goat anti-mouse and anti-rabbit secondary antibodies (Sigma-Aldrich, Saint Louis, MO, USA) and analyzed using a flow cytometer. The protein levels of the integrin α2 and α4 subunits and αVβ3 complex were quantified using FCS Express 4 software (Becton Dickinson, CA, USA).

#### 4.4.2. ARS Staining

Intracellular calcium mineralization formed by osteocyte differentiation was observed through ARS (Sigma-Aldrich, Saint Louis, MO, USA) staining. First, 1 × 10^4^ MSCs were seeded onto each material and cultured at 37 °C in a 5% CO_2_ incubator for 3, 5, 7, or 10 days. Next, the cells were washed twice with PBS, fixed with 4% PFA for 15 min, and rewashed with PBS to remove the remaining PFA. Then, a 2% ARS staining solution was prepared by dissolving 800 mg ARS powder in 40 mL of deionized H_2_O, filtering it, and then adjusting its pH to 4.1–4.3. The cells were incubated in 500 μL of the ARS staining solution for 15 min, followed by the same volume of deionized H_2_.

#### 4.4.3. ORO Staining

Adipocyte differentiation was observed using the lipophilic dye ORO (Sigma-Aldrich, Saint Louis, MO, USA). First, 1.4 g of ORO powder was dissolved in 400 mL of 99% isopropanol at room temperature for 12 h, filtered, mixed with 144 mL of deionized H_2_O, then placed in a 4 °C refrigerator for 12 h. Next, it was filtered again and left to stand for 30 min to form the ORO dye. Then, 1 × 10^4^ MSCs were seeded onto each material and cultured at 37 °C in a 5% CO_2_ incubator for 3, 5, 7, or 10 days. Next, the cells were washed twice with PBS, fixed with 4% PFA for 15 min, rinsed with 60% isopropanol for 30 s, and then separately incubated with 1 mL of ORO dye and Mayer’s hematoxylin for 10 min each. Finally, the MSCs were washed with deionized H_2_O twice, dried at room temperature, and observed under a microscope.

### 4.5. Animal Models

The in vivo biocompatibility assessments used 2–3-month-old female SD rats weighing 300–350 g. The study protocol was approved by the Animal Care and Use Committee (approval number La-1071565). The materials (TCPS, FN, FN-Au, Col, and Col-Au) were implanted locally under the dorsal skin through a 10 mm incision under local anesthesia. After one month, capsule formation was observed through H&E staining, and collagen deposition was measured using Masson’s trichrome staining. Macrophage activation was investigated by adding mouse monoclonal anti-CD86 and anti-CD163 antibodies (Cell Signaling, MA, USA) diluted 1:200. Additionally, endothelial differentiation was investigated by adding an anti-CD31 antibody (Santa Cruz Biotechnology, Santa Cruz, CA, USA). These experimental results are presented as the mean ± standard deviation (SD) of five rats.

### 4.6. Statistical Analysis

The data were statistically analyzed using SPSS Statistics software (version 17.0). The results for each experiment are presented as the mean ± SD. We ensured reproducibility by independently repeating all experiments three times. The difference between groups was evaluated using Student’s *t*-test. A *p*-value of <0.05 was considered statistically significant: *, *p* < 0.05; **, *p* < 0.005; ***, *p* < 0.001 compared to the control group (TCPS); ^##^, *p* < 0.01; ^###^, *p* < 0.001 compared to the FN or Col group).

## 5. Conclusions

This study successfully fabricated FN-Au and Col-Au nanocomposites (12.2 ppm) and demonstrated their excellent dispersity and even distribution as a thin layer suitable for cell growth, which may relate to the creation of a better nanotopography, as well as possibility due to the optimal concentration of AuNPs, which cause the better dispersity property. The biocompatibility and cell function results showed that the FN-Au and Col-Au nanocomposite coatings compare to pure FN and Col, which refer to potentially improving the biocompatibility of clinical biomaterials. In addition, they also improved MMP2/9 expression and promoted MSC proliferation and migration. Moreover, the gene expression, IF staining, ARS staining, and ORO staining indicated that these materials significantly induced MSCs to differentiate into neurons, osteoblasts, adipocytes, and endothelial cells. Furthermore, capsule formation, collagen deposition, CD86/163 expression, and CD31 expression in the animal experiments also indicated the excellent biocompatibility of the FN-Au and Col-Au nanocomposites. In conclusion, this study proved that adding AuNPs to biomolecules (FN and Col) results in more biostable, biocompatible materials and can significantly improve stem cell differentiation, making it a promising material for future clinical use.

## Figures and Tables

**Figure 1 ijms-25-07241-f001:**
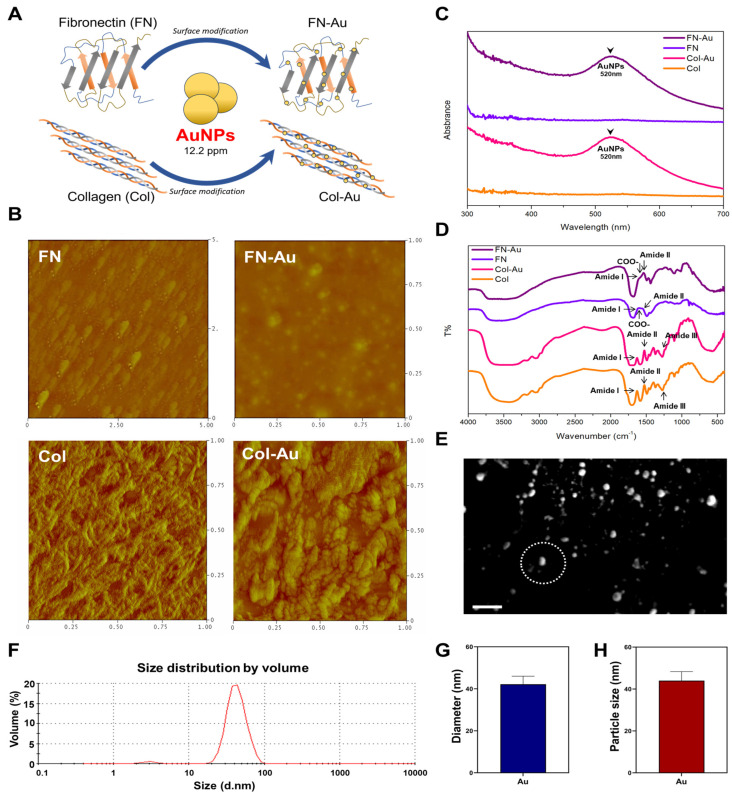
Materials preparation and surface characterization of nanocomposites. (**A**) Brief scheme for preparing different nanocomposites with 12.2 ppm AuNPs, including FN, FN-Au, Col, and Col-Au. (**B**) AFM topography diagrams for FN, FN-Au, Col, and Col-Au. (**C**) UV-Vis spectra. The peak at 520 nm was found in FN-Au and Col-Au, proving the successful preparation process. (**D**) FTIR spectra, with functional groups labeled on the specific peaks. An amide III band was found in FN-Au and Col-Au groups. (**E**) SEM images of gold nanoparticles. Scale bar = 200 nm (**F**) Size distribution graph obtained from the DLS assay. The mean and SD of the (**G**) particle size and (**H**) diameter were also recorded.

**Figure 2 ijms-25-07241-f002:**
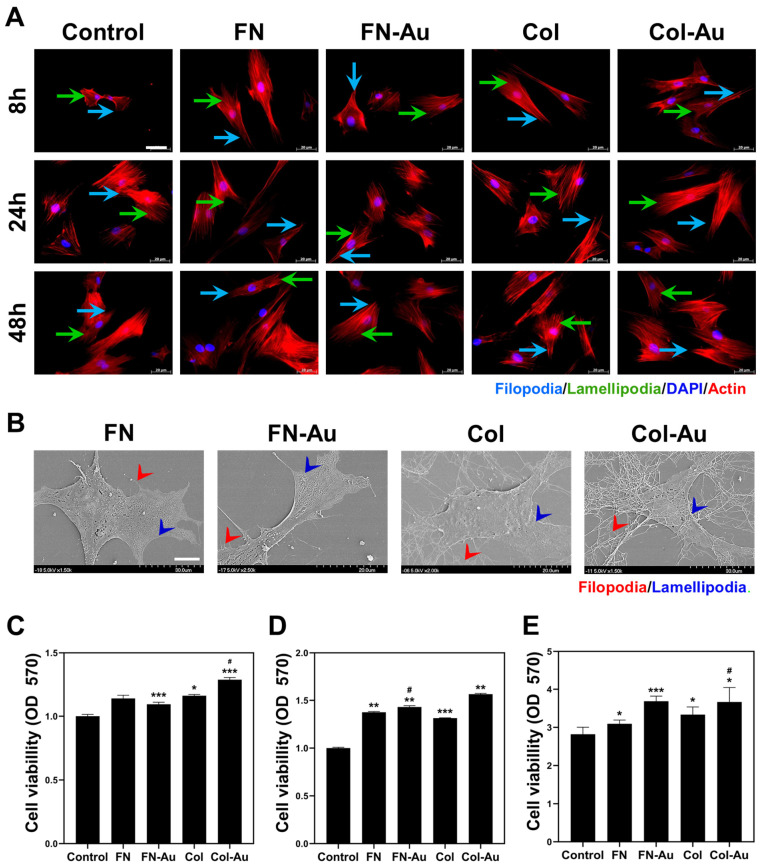
Cell morphology and cell viability analysis. (**A**) Rhodamine-phalloidin staining for the actin fibers on the control, FN, FN-Au, Col, and Col-Au for 8, 24, and 48 h, respectively. The scale bar is 20 μm. (Blue arrow: filopodia, green arrow: lamellipodia, blue color: nucleus (DAPI staining), red color: actin fiber) Scale bar = 20 μm. (**B**) SEM images for MSCs on FN, FN-Au, Col, and Col-Au at 48 h. Scale bar = 5 μm (**C**–**E**) MSC viability on each material for 24, 48, and 72 h, respectively. All results show significant viability in FN-Au and Col-Au when compared to the control and the two pure substances. * *p* < 0.05, ** *p* < 0.01, *** *p* < 0.001: compared to the control. # *p* < 0.05: compared to the pure substances (FN and Col).

**Figure 3 ijms-25-07241-f003:**
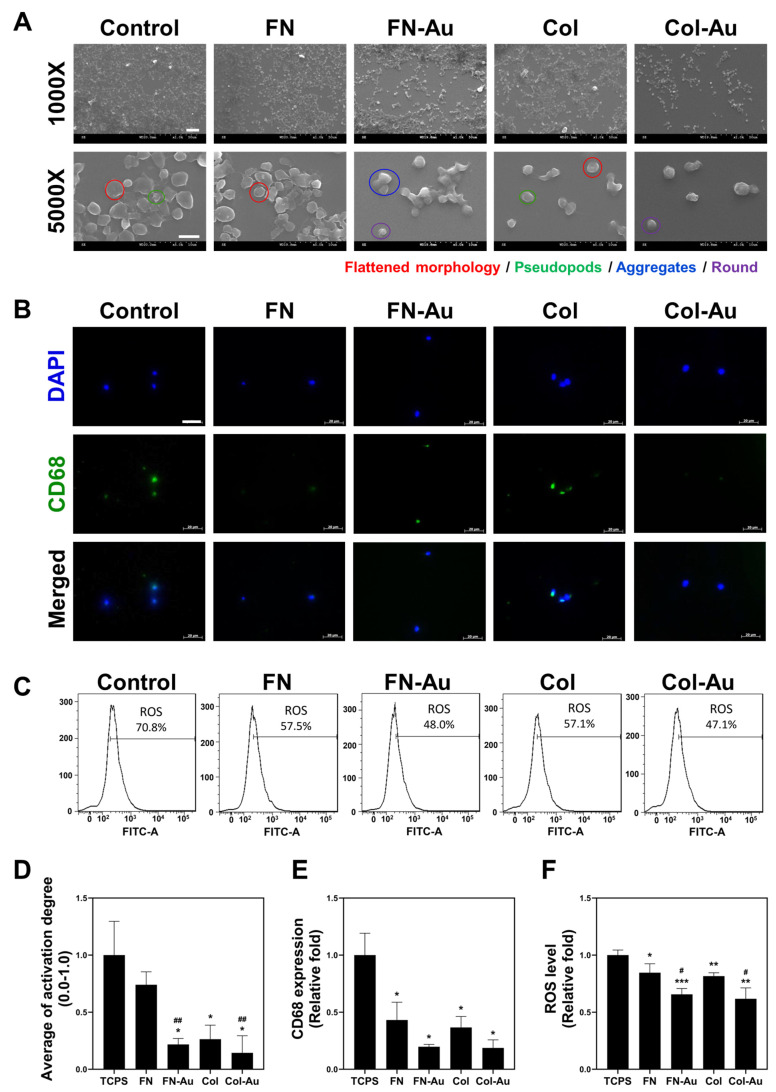
In vitro biocompatibility assay of different materials. (**A**) Platelet morphology was observed using SEM on different materials, and each type of appearance is labeled, including flattened (red), pseudopods (green), aggregates (blue), and round (purple), ranging from the activated to inactivated form. Scale bar = 10 μm (1000×) and scale bar = 5 μm (5000×). (**B**) The expression of CD68 in macrophages on different materials at 96 hr. CD68 macrophage marker expression was exhibited using IF staining (green) with the nucleus stained with DAPI (blue). Scale bar = 20 µm. (**C**) The intracellular reactive oxygen species (ROS) quantified using 2,7-dichlorofluorescein diacetate (DCFH-dA) and flow cytometric analysis of MSCs on different materials. ROS generation was detected using the FACS methods, and the results showed lower ROS generation in the two nanocomposite groups. (**D**–**F**) Semi-quantification results of the platelet activation degree, CD68 level, and ROS level, respectively. Note that the average platelet activation degree is shown in a range between 0 and 1.0, while the others are shown as the relative fold in order to compare them simply. * *p* < 0.05, ** *p* < 0.01, *** *p* < 0.001: compared to the control. # *p* < 0.05, ## *p* < 0.01: compared to the pure substances (FN and Col).

**Figure 4 ijms-25-07241-f004:**
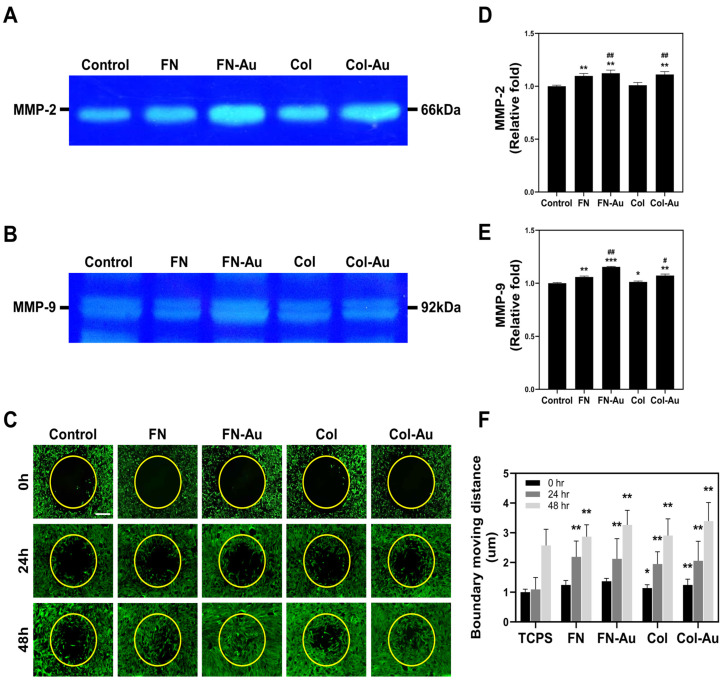
MMP-2/9 zymography and cell migration assay. (**A**) MMP-2 and (**B**) MMP-9 zymography was conducted to compare the expression between each group and to evaluate their enzymatic activity. (**C**) The migration of MSCs was observed in different groups after 0, 24, and 48 h. MSCs were stained with Calcein-AM and then monitored via fluorescence microscopy. Scale bar = 200 μm. (**D**,**E**) The semi-quantitative results are based on a detecting of the optical density of the bands, and both of them show significantly higher expression on FN-Au and Col-Au. (**F**) The migration distance of MSCs on different materials was also semi-quantified, and the data were expressed as the mean ± SD (*n* = 3). * *p* < 0.05, ** *p* < 0.01, *** *p* < 0.001: compared to the control. # *p* < 0.05, ## *p* < 0.01: compared to the pure substances (FN and Col).

**Figure 5 ijms-25-07241-f005:**
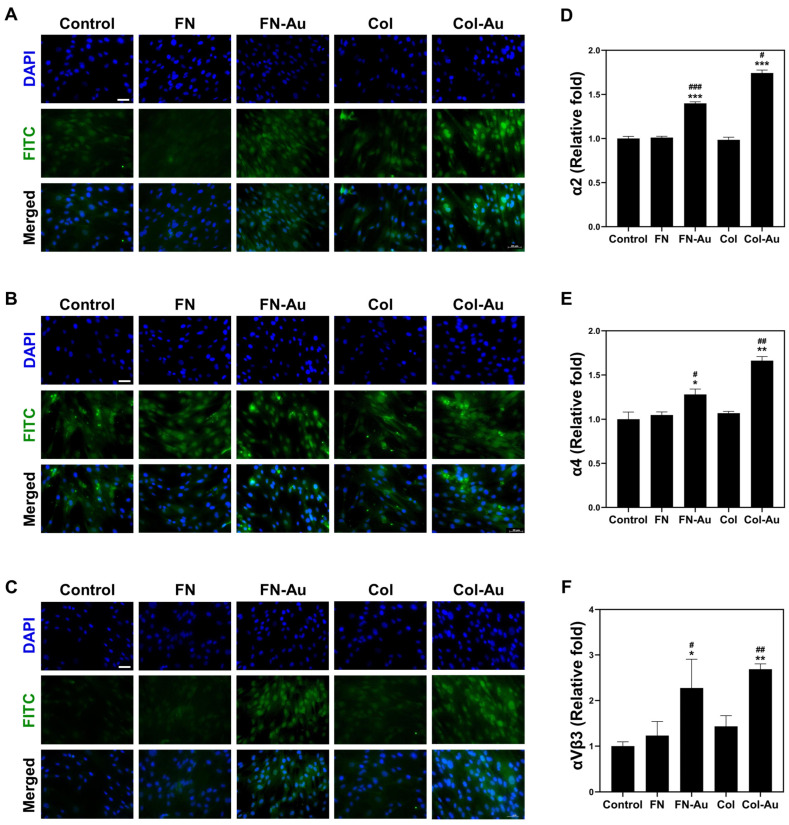
α2, α4, and αVβ3 integrin expression was observed using a fluorescent microscope in each group. The cells were immunostained with α2, α4, and αVβ3 antibodies and then conjugated with FITC-immunoglobulin secondary antibodies (green), and the nucleus was again stained with DAPI (blue). The images were displayed as (**A**) α2, (**B**) α4, and (**C**) αVβ3, respectively. Scale bar = 20 μm. (**D**–**F**) The quantified results of α2, α4, and αVβ3 integrin expression, respectively. The results show that α2, α4, and αVβ3 expression are maximum in the FN-Au and Col-Au groups. * *p* < 0.05, ** *p* < 0.01, *** *p* < 0.001: compared to the control. # *p* < 0.05, ## *p* < 0.01, ### *p* < 0.001: compared to the pure substances (FN and Col).

**Figure 6 ijms-25-07241-f006:**
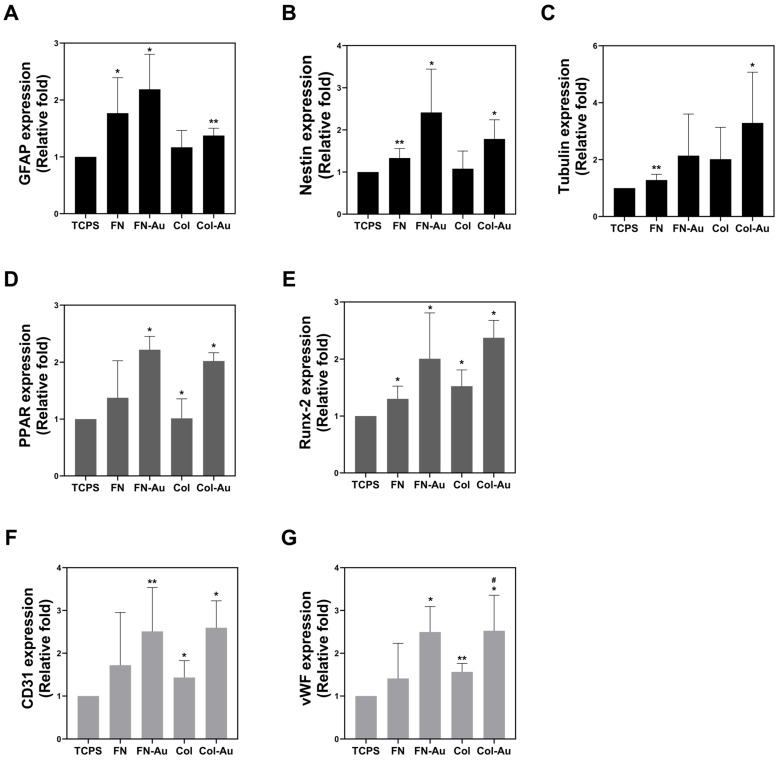
Gene expression of the seven differentiation markers in MSCs was determined via a real-time PCR technique. (**A**–**C**) The expression of the GFAP, Nestin, and tubulin neural differentiation markers was semi-quantified. (**D**) The expression of the PPAR adipocyte differentiation marker was semi-quantified. (**E**) The expression of the Runx-2 osteoblastic differentiation marker was semi-quantified. (**F**,**G**) The expression of vWF and CD31 endothelial differentiation markers was also semi-quantified. * *p* < 0.05, ** *p* < 0.01: compared to the control. # *p* < 0.05: compared to the pure substances (FN and Col).

**Figure 7 ijms-25-07241-f007:**
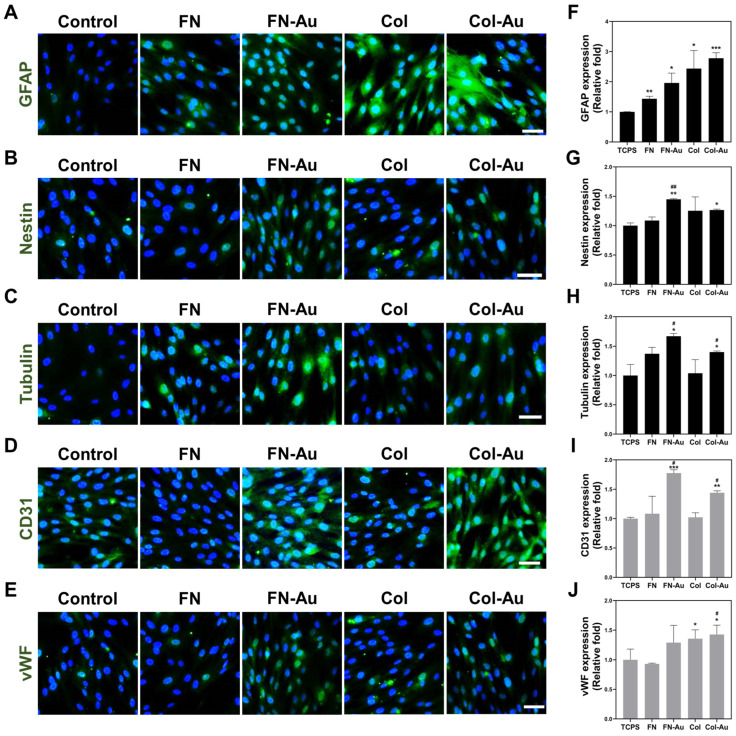
The multiple-differentiation capability of MSCs in various materials. (**A**–**E**) The MSCs were stained with five primary antibodies and then conjugated with secondary FITC-immunoglobin secondary antibodies. The images of day 3 and day 5 are shown in the supplementary documents. Scale bar = 20 μm. (**F**–**H**) The expression of GFAP, Nestin, and tubulin was used to assess neural differentiation. (**I**,**J**) vWF and CD31 endothelial markers were also semi-quantified to assess endothelial differentiation. * *p* < 0.05, ** *p* < 0.01, *** *p* < 0.001: compared to the control. # *p* < 0.05, ## *p* < 0.01: compared to the pure substances (FN and Col).

**Figure 8 ijms-25-07241-f008:**
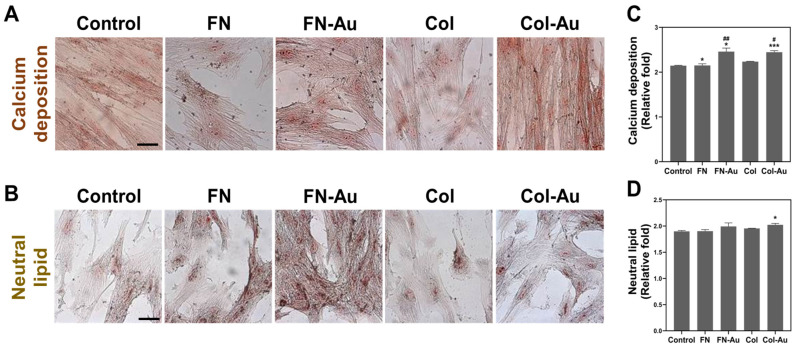
Further investigation of osteoblast and adipocyte differentiation of MSCs using ARS and ORO staining at day 7. (**A**) Calcium deposition of MSCs was stained with ARS staining. Scale bar = 20 μm. (**B**) The neutral lipids in the MSCs were stained with ORO staining. Scale bar = 20 μm. (**C**,**D**) The semi-quantitative results for calcium deposition and neutral lipids. * *p* < 0.05, *** *p* < 0.001: compared to the control. # *p* < 0.05, ## *p* < 0.01: compared to the pure substances (FN and Col).

**Figure 9 ijms-25-07241-f009:**
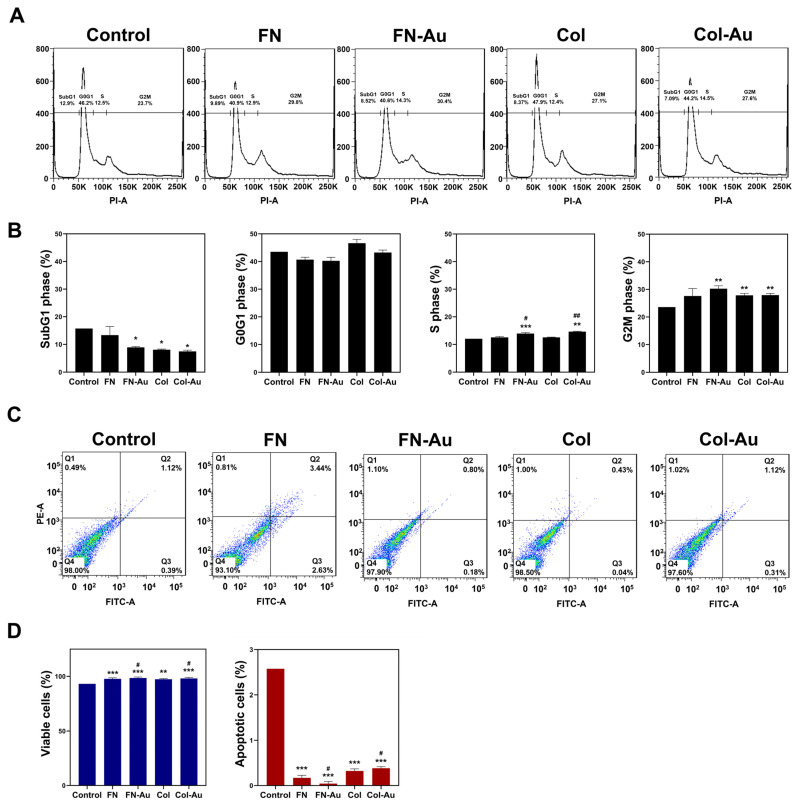
The cell cycle progression and the apoptotic rate of MSCs on various materials. (**A**,**B**) MSCs were stained with PI, and the DNA was examined via flow cytometry; 104 cells were counted in each material, and the data were computed using FACS software. (**C**) The cells were co-stained with PI and annexin V, and the apoptotic MSCs were investigated via flow cytometry. (**D**) The semi-quantitative data shows the number of viable and apoptotic cells presented as the mean ± SD (%). * *p* < 0.05, ** *p* < 0.01, *** *p* < 0.001: compared to the control. # *p* < 0.05, ## *p* < 0.01: compared to the pure substances (FN and Col).

**Figure 10 ijms-25-07241-f010:**
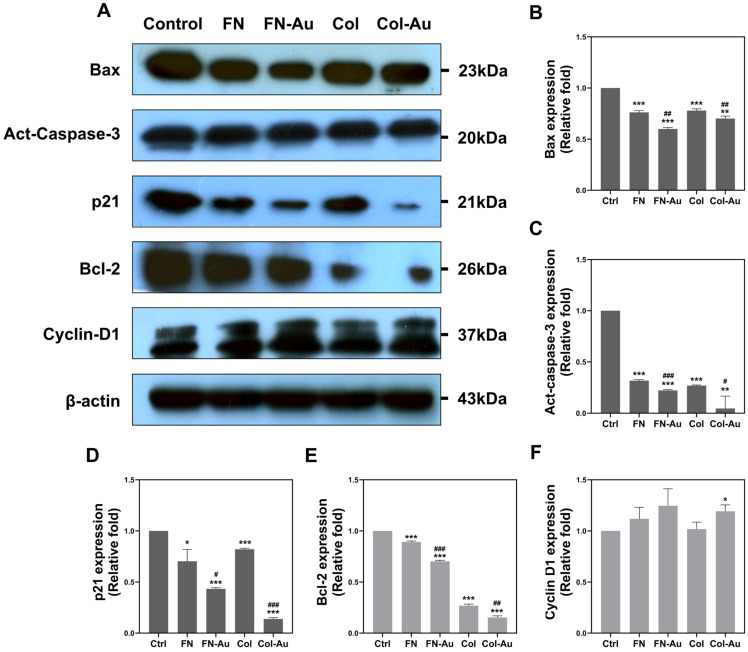
The expression of pro-apoptotic and anti-apoptotic proteins on various materials. (**A**) Western blotting (WB) assay with primary anti-Bax, anti-act-caspase-3, anti-p21, anti-Bcl-2, and anti-cyclin D1 antibodies, and β-actin was used as the control in this experiment. (**B**–**F**) The expression of apoptotic proteins (Bax, act-caspase-3, and p21) and anti-apoptotic proteins (Bcl-2 and Cyclin D1) was then semi-quantified. Data are expressed as the mean ± SD (*n* = 3). * *p* < 0.05, ** *p* < 0.01, *** *p* < 0.001: compared to the control. # *p* < 0.05, ## *p* < 0.01, ### *p* < 0.001: compared to the pure substances (FN and Col).

**Figure 11 ijms-25-07241-f011:**
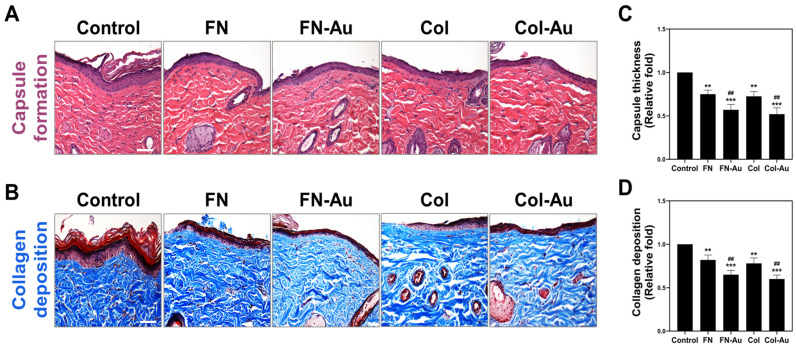
Biocompatibility investigations in the rat model, including the measurement of capsule thickness and the deposition of collagen. The tissue slices were derived from the female SD rats. (**A**) Capsule formation was stained with H&E. Scale bar = 200 μm. (**B**) Collagen deposition was stained with Masson’s trichrome staining assay (blue area: collagen fibers). Scale bar = 200 μm. (**C,D**) The above images of the control, FN, FN-Au, Col, and Col-Au were all semi-quantified in the bar graph shown beside. ** *p* < 0.01, *** *p* < 0.001: compared to the control. ## *p* < 0.01: compared to the pure substances.

**Figure 12 ijms-25-07241-f012:**
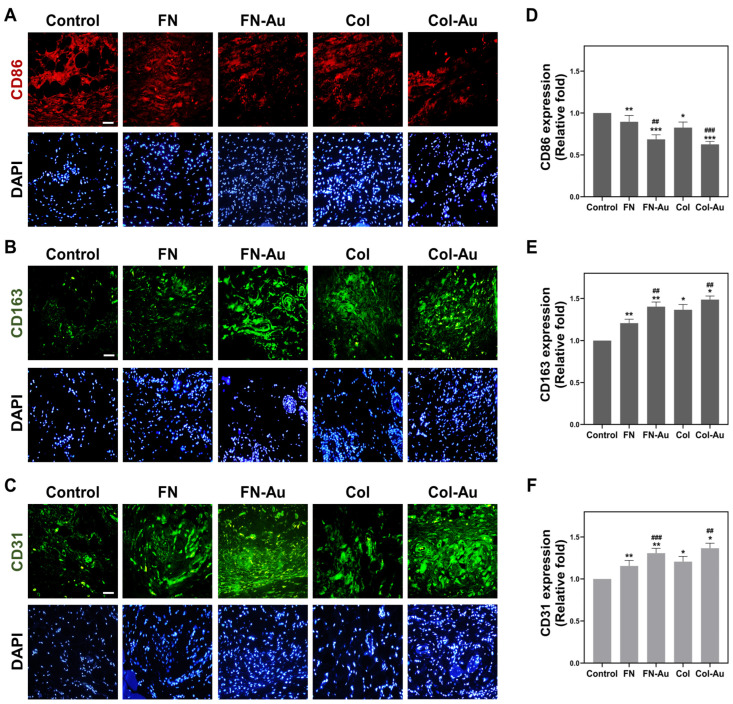
Evaluation of anti-inflammatory capabilities and endothelial markers in rat models. (**A**) CD86, the specific marker of M1 macrophages, was stained in red with the nucleus stained with DAPI (blue). (**B**) CD163, the specific marker of M2 macrophages, was stained in green with the nucleus stained with DAPI (blue) again. (**C**) CD31, also known as the endoCAM, was used as the endothelial marker (green) for this experiment. Scale bar = 50 μm. (**D**–**F**) The semi-quantified results of CD86, CD163, and CD31, sequentially. * *p* < 0.05, ** *p* < 0.01, *** *p* < 0.001: compared to the control. ## *p* < 0.01, ### *p* < 0.001: compared to the pure substances.

**Figure 13 ijms-25-07241-f013:**
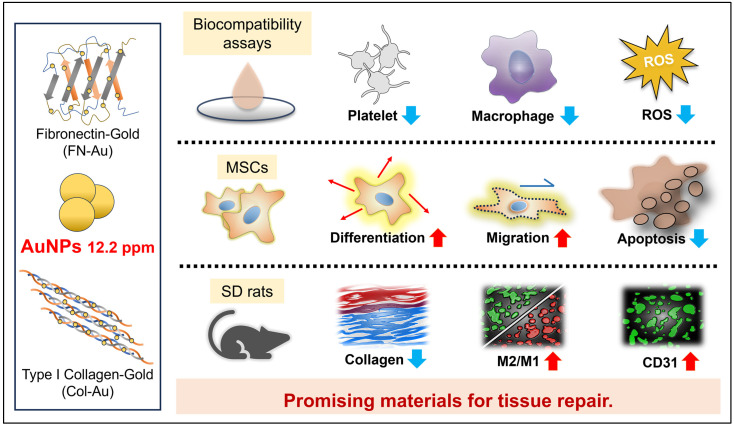
Scheme for experimental abstract. Of note, the FN-Au (12.2 ppm) and Col-Au (12.2 ppm) nanocomposites have better biocompatibility than FN and Col both in vitro and in vivo (SD rat models), and they can also significantly improve MSC multiple differentiation, including neural, bone, adipocyte, and endothelial differentiation.

**Table 1 ijms-25-07241-t001:** Quantification data of MSC morphology for 8, 24, and 48 h: (**A**) size (μm) and (**B**) area (μm^2^).

**A. Size (μm)**			
**Materials**	**8 h**	**24 h**	**48 h**
Control (TCPS)	10.7 ± 1.2	12.7 ± 0.6	15.3 ± 0.6
FN	15.3 ± 1.5 *	17.0 ± 1.0 *	17.7 ± 1.5
FN-Au	21.0 ± 1.0 **^,##^	22.3 + 2.1 **^,#^	23.7 ± 1.5 **^,##^
Col	14.3 ± 1.5 **	15.7 ± 0.6 *	16.7 ± 0.6 *
Col-Au	18.0 + 2.0 *	17.3 ± 0.6 **^,#^	22.0 ± 2.0 **^,#^
**B. Area (μm^2^)**			
**Materials**	**8 h**	**24 h**	**48 h**
Control (TCPS)	22.0 ± 1.0	24.7 ± 0.6	28.3 ± 0.6
FN	23.7 ± 1.5	25.7 ± 1.5	30.7 ± 0.6 **
FN-Au	26.3 ± 1.2 *^,##^	28.7 ± 0.6 *	34.3 ± 1.5 *^,#^
Col	23.7 ± 0.6	25.7 ± 1.2	28.0 ± 1.0
Col-Au	25.3 ± 0.6 *^,#^	28.3 ± 1.2 *^,#^	32.3 ± 1.5 *^,#^

* *p* < 0.05, ** *p* < 0.01: compared to the control. # *p* < 0.05, ## *p* < 0.01: compared to the pure substances (FN and Col).

**Table 2 ijms-25-07241-t002:** The conversion yield (%) of monocytes to macrophages on various materials was calculated after 96 h of culture, and the data were presented as the mean ± SD.

Materials	The Number of Monocytes (× 10^4^)	The Number of Macrophages (× 10^4^)	Conversion (%)
**Control (TCPS)**	52 ± 4.00	11.3 ± 2.08	17.9 ± 0.19
**FN**	46 ± 4.50	4.3 ± 1.52	8.54 ± 0.15 **
**FN-Au**	53 ± 8.50	2.3 ± 0.57	4.16 ± 0.01 ***^,##^
**Col**	51 ± 8.00	4.0 ± 1.00	7.27 ± 0.09 ***
**Col-Au**	55 ± 5.68	2.3 ± 1.15	4.01 ± 0.07 ***

** *p* < 0.01, *** *p* < 0.001: compared to the control. ## *p* < 0.01: compared to the pure substances (FN and Col).

## Data Availability

Data are contained within the article and Appendix A.

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
