# Peer review of "Assessment of the Biocompatibility Ability and Differentiation Capacity of Mesenchymal Stem Cells on Biopolymer/Gold Nanocomposites"

_ijms, 2024, doi:10.3390/ijms25137241_

Round 1

Reviewer 1 Report

Comments and Suggestions for Authors

The above mentioned paper is well organized and includes a lot of data which, presumably, would have been sufficient to prepare two papers. Concerning the biological response of the different types of cells I suggest to assemble all the results obtained in  MSC and to better address the legend to the figures. For example legend to fig.3 is quite difficult to read since I don't understand why authors show platelets ROS production instead of MSC ROS production. In addition  in results and discussion sections is overall reported that MSC have a better proliferation or differentiation effect  and so on instead of AunP and ColAu have a proliferative or differentiative effect: the effect is due to the material, cells undergo the effect. 

Comments on the Quality of English Language

Minor revision

Author Response

It is very much appreciated that the reviewers made these valuable comments. We have attended to the comments and made the revision accordingly.

Response to Reviewer 1 Comments:

Comments and Suggestions for Authors:

  1. The above mentioned paper is well organized and includes a lot of data which, presumably, would have been sufficient to prepare two papers.

Response:

We appreciate and thanks for the comment from reviewer.

  1. Concerning the biological response of the different types of cells I suggest to assemble all the results obtained in MSC and to better address the legend to the figures. For example legend to fig.3 is quite difficult to read since I don't understand why authors show platelets ROS production instead of MSC ROS production.

Response: 

Thanks for the valuable comment. We have revised the wording of Figure 3 in the “Figure caption” section and the “Results” section make it to be easy follow for readers. 

  • “Figure caption” section:

“Figure 3. In vitro biocompatibility assay of different materials. (A) Platelet morphology was observed using SEM on different materials, and each type of appearance is labeled, including flattened (red), pseudopods (green), aggregates (blue), and round (purple), ranging from activated to inactivated form. (B) The expression of CD68 for macrophages on different materials at 96 hr. CD68 macrophage marker expression was exhibited using IF staining (green) with the nucleus stained with DAPI (blue). (Scale bar=20µm) (C) The intracellular reactive oxygen species (ROS) quantified by 2,7-dichlorofluorescein diacetate (DCFH-dA) and flow cytometric analysis of MSCs on different materials. ROS generation was detected using the FACS methods, and the results showed lower ROS generation in the two nanocomposite groups. (D-F) Semi-quantification results of platelet activation degree, CD68 level, and ROS level, respectively. Note that the average platelet activation degree is shown in a range between 0-1.0, while the others are shown in the relative fold in order to compare them simply. *p < 0.05, **p < 0.01, ***p < 0.001: compared to the control. #p<0.05, ##p<0.01, ###p<0.001: compared to the pure substances (FN and Col).” (Line 283-292)

  • “Results” section:
  • “MSC cells” (Line 307)
  • “MSC cells” (Line 317)

  1. In addition in results and discussion sections is overall reported that MSC have a better proliferation or differentiation effect and so on instead of AuNP and ColAu have a proliferative or differentiative effect: the effect is due to the material, cells undergo the effect. 

Response:

Thanks for the value comment from the reviewer. We have included the more detail description in the “Discussion” section. “This study intended to provide a functionalized surface coating comprised of AuNPs and FN or Col to improve the biocompatibility and favorable cellular response of MSCs on the nanocomposites. The surface morphology of FN and Col was markedly changed by adding a small amount of AuNPs. The FN-Au and Col-Au nanocomposites exhibited better biocompatibility and biological performance than the pure FN and Col, including promoting cell proliferation, reducing platelet and monocyte activation, and reducing ROS generation. In addition, the FN-Au and Col-Au nanocomposites induced better MSC differentiation, making them promising materials for tissue regeneration.” (Line 613-620) 

Reviewer 2 Report

Comments and Suggestions for Authors

The work is well presented, but there are major problems to solve. First of all the writing is very poor and must be thoroughly improved.

Line 28: ''two types of nanogold composite were observed''...this is not the correct verb.

Line 73: ''Our lab'', change this expression.

Line 104, please revise this expression: ''forms depending on the situation''

Line 167-168. Both sentences start with ''This study''.

Line 169, what does ''Vitro cells'' mean?

In the final part of the introduction, can you clarify what does this study want to explore? What will be the final application of the gold nanoparticles?

Method

Dscribe the method indirectly. You shouldn't say ''we filtered, we added etc''. Line 189: ''after mixing well'' is not very scientific. 

Line 197, revise the syntax.

Paragraph 2.1.3 the syntax must be revised.

Line 209, you don't need to say how the techniques work.

Paragraph 2.2.1, please explain how the MSC were isolated from the umbelical cord and show the CD marker characterization. 

You must specify, the city and the country of production for all the reagents. 

'' 2x105 cells'' check the writing.

''After that, remove the supernatant, add an- 237 other culture medium without 10% bovine serum albumin (BSA) for 24 hours, and ob- 238 serve the differentiation of MSCs on these plate''. This sentence needs revision of the syntax.

2.2.2 needs revision. ELISA reader? or plate reader? Line 248, were the nanoparticles tested in complete cell culture media, please give details.

2.2.3. How was the coating successfully obtained?

2.2.4 What was the age of the volunteer?

''To extract intracellular total RNA, use TRIZOL''...it seems like you have copied and pasted the methods from a lab book, without checking the correctness of the wiriting. Carefully revise the manuscript.

For the conclusions, you have said you have demonstrated excellent ''dispersity''. How did you measure it?

''FN-Au and Col-Au have a significantly lower inflammatory response, plate- 914 let activation degree, apoptotic rate, and ROS generation'' is this compared to what?

For the ORO and ARS staining, positive controls should be included to compare the figures with the appearance of true positives.

Comments on the Quality of English Language

Extensive revision of the writing is needed. It seems as the authors didn't take time to carefully prepare the manuscript's method part, which appears it has been simply copied and pasted from an internal protocol.

Author Response

Ms. Ref. No: ijms-3023674

Title: Assessment of the Biocompatibility Ability and Differentiation Capacity of Mesenchymal Stem Cells on Biopolymer/Gold Nanocomposites

It is very much appreciated that the reviewers made these valuable comments. We have attended to the comments and made the revision accordingly.

Response to Reviewer 2 Comments:

Comments and Suggestions for Authors

The work is well presented, but there are major problems to solve. First of all the writing is very poor and must be thoroughly improved.

Response:

Thanks for the valuable comment from the reviewer. We have carefully addressed and modified the writing following your kindness suggestion.

  1. Line 28: ''two types of nanogold composite were observed''...this is not the correct verb.

Response:

Thanks for the valuable comment from the reviewer. We have corrected for this sentence to “This study assessed the biocompatibility of two types of nanogold composite: fibronectin-gold (FN-Au) and collagen-gold (Col-Au).” (Line 30-31)

  1. Line 73: ''Our lab'', change this expression.

Response:

We have modify of “Our lab” to “Our group” (Line 77)

  1. Line 104, please revise this expression: ''forms depending on the situation''

Response:

We have revised the expression “FN is a dimeric protein that can exist in different forms depending on the situation and has a molecular weight of around 500 kDa [18].” (Line 108-109)

  1. Line 167-168. Both sentences start with ''This study''.

Response:

We have modified this sentence to “Therefore, this study used AuNPs to modify FN and Col, creating FN-Au and Col-Au nanocomposites, and assessed their properties using pluripotent MSCs, which are widely used clinically since they can differentiate into various cell types.” (Line 164-167)

  1. Line 169, what does ''Vitro cells'' mean?

Response:

This is a typo. We have corrected to “MSCs can be cultured and attached to the bottom of plates in vitro, forming fibroblast colony-forming units.” (Line 167-168)

  1. In the final part of the introduction, can you clarify what does this study want to explore? What will be the final application of the gold nanoparticles?

Response:

We have modified for this sentence to “This study examined the effects of surface modification of FN and Col by AuNPs, which may enhance the biological performance and differentiation capacity of MSCs. Therefore, it explored whether the FN-Au and Col-Au nanocomposites show promise for fabricating the surface of biomedical devices.” (Line 178-181)

Method:

  1. Dscribe the method indirectly. You shouldn't say ''we filtered, we added etc''. Line 189: ''' is not very scientific. 

Response: 

  • We have modified to “The AuNPs” (Line 641)
  • We have modified to “were added” (Line 642)

  1. Line 197, revise the syntax.

Response:

We have revised the sentence to “to observe the morphology of MSCs” (Line 653).

  1. Paragraph 2.1.3 the syntax must be revised.

Response:

We have revised the sentence to “4.1.3. Fourier-Transform Infrared Spectroscopy analysis (FTIR)” (Line 657)

  1. Line 209, you don't need to say how the techniques work.

Response:

We have deleted for this sentence.

  1. Paragraph 2.2.1, please explain how the MSC were isolated from the umbelical cord and show the CD marker characterization. 

Response:

(1) We have included the more detail description of “Human umbilical cord (Wharton’s jelly)-derived mesenchymal stem cells (MSCs) was kindly provided by Prof. Woei-Cherng Shyu (China Medial University, Taiwan)” (Line 681-682)

(2) We have provided the new CD markers characterization of MSC in the new Figure S1 and description in the:

1) “Materials and Methods” section:

“The specific surface marker of the MSCs were characterized by flow cytometry. MSCs were incubated with antibodies conjugated with Fluorescein Isothiocyanate (FITC) and Phycoerythrin (PE) using the markers: CD14-FITC, CD45-FITC, CD44-PE, and CD73-PE (BD Pharmingen, CA, USA). Isotype controls were demonstrated by PE-conjugated IgG1 and FITC-conjugated IgG1 (BD Pharmingen, San Diego, CA, USA). FACS software (Becton Dickinson LSR II, MA, USA) was used to analyze the phenotype of the MSCs.” (Line 696-671)

  • “Results” section

“The negative markers CD14 and CD45 were highly expressed in hematopoietic and immune cells, respectively. The positive markers CD44 and CD73 for MSCs were significantly expressed based on the flow cytometry analysis (Figure S1).” (Line 221-223)

  • “Figure caption” section

“Phenotype characterization of MSCs. (A) The antibodies were conjugated with Fluorescein Isothiocyanate (FITC) and Phycoerythrin (PE), with the following markers of CD14-FITC, CD45-FITC, CD44-PE, and CD73-PE and analyzed by flow cytometry. All the results are representative of one of three independent experiments.” (Supplemental Figure S1)

  1. You must specify, the city and the country of production for all the reagents. 

Response:

We have included the city and the country for all the reagents in this manuscript following reviewer’s kindness suggestion.

  1. '' 2x105 cells'' check the writing.

Response:

We have modified the sentence to “Cells at a density of 2x105 cells per well were seed” (Line 692)

  1. ''After that, remove the supernatant, add an- 237 other culture medium without 10% bovine serum albumin (BSA) for 24 hours, and ob- 238 serve the differentiation of MSCs on these plate''. This sentence needs revision of the syntax.

Response:

We have modified the sentence to “Afterwards, the supernatant was removed and replaced with 10% bovine serum albumin-free culture for 24 hours, and then the differentiation of MSCs on these plates was observed.” (Line 694-696)

  1. 2.2.2 needs revision. ELISA reader? or plate reader? Line 248, were the nanoparticles tested in complete cell culture media, please give details.

Response:

We have revised for this section to t “The of materials (TCPS, FN, FN-Au, Col, and Col-Au) on cell growth was determined by MTT assay. Cells (200 μl/well at a density of 2×104 cells/mL) were cultivated with medium containing test samples in a 96-well culture plate at 37°C with 5% CO2. After 24, 48 and 72 hours incubating, the medium is removed and MTT solution, 100 μL (0.5 mg/mL), was added into each well, and the plate was placed in incubation room for at 37°C for 2-4 hours. After then, MTT solution was removed, and 100 μL DMSO is added for 15-30 minutes. An ELISA reader is finally used to detect the 570 nm OD value of each group. Then cell growth was determined by measuring the absorbance at 570 nm. Cell growth rate (%) was assessed using the following formula: (%) = [(OD1-OD0)/(OD2-OD0)] ×100%, where OD0, OD1, and OD2 represent the mean OD of the blank, experimental, and control groups, respectively. All experiments were performed in triplicate.” (Line 703-712)

  1. 2.3. How was the coating successfully obtained?

Response:

Thanks for the valuable comment from the reviewer. We have modified the description for this section. “Materials were coating on coverslip glass and were placed in a 24-well culture plate and 0.5 ml of platelet-rich plasma (2x106 platelets/ml). After then, samples were removed after incubation for 1 hour and the number of adherent platelets were counted by a cell counter (Assistant, Konigswinter, Germany).” (Line 714-717)

  1. 2.4 What was the age of the volunteer?

Response:

Thanks for the valuable comment from the reviewer. Platelet was obtained from health volunteer. Because the samples are provided by the blood bank center of TVGH Blood Center, we cannot to know the age of the volunteer regarding on the law of personal data protection.

  1. ''To extract intracellular total RNA, use TRIZOL''...it seems like you have copied and pasted the methods from a lab book, without checking the correctness of the wiriting. Carefully revise the manuscript.

Response:

We have carefully revised for this section. “The RNA expression level of MSCs was extracted by TRIzol (lnvitrogen, MA, USA), and the experiments were applied and following the protocol of manufacturer’s instructions. The 2 x105 MSCs per well were cultured in 6-well culture plates with the different materials coatings. The cells were incubated for 3, 5, and 7 days under cell culture condition (37 oC, 5% CO2). After incubation, the medium was removed, and TRIzol solution (1 mL) was added into each well for 5 min to collect the cells. Afterwards, 200 mL of chloroform (Sigma-Aldrich, Saint Louis, USA) was added into each well for 15 second, and the culture plates were left for 3 minute at room tempature. Next, the samples were collected for 12,000 rpm, 15 min, 4oC centrifugation. Afterward, the supernatant was discarded, and isopropanol (500 mL) was loaded (4oC, 10 min), followed by centrifugation for 15 min (12,000 rpm, 4oC). The samples were washed two times with 75% alcohol (1 mL) after discarding the supernatant. Next, samples containing RNA were dried out. Next, 20 mL of DEPC-treated H2O was loaded into the samples, while the data were analyzed at 260 nm by a ELISA reader (SpectraMax M2, Molecular Devices, SanJose, CA, USA). RevertAidTM First Strand cDNA Synthesis Kit (Fermentas, Burlington, Canada) was applied for cDNA synthesis following by the manufacturer’s instructions. The RNA expression in MSCs with various treatments was identified by Step OneTM Plus Real-Time PCR System” (Line 784-800)

  1. For the conclusions, you have said you have demonstrated excellent ''dispersity''. How did you measure it?

Response:

1) Thanks for the valuable comments from the reviewer. We have included the description in the “Conclusion” section. “which may relate to create a better nanotopography as well as possibility due to the optimal concentration of AuNPs which cause the better dispersity property.” (Line 928-929)

2) We can measure the surface roughness (Ra) of the FN-Au and Col-Au and further evaluate its dispersion property by AFM analysis results (Figure 1B).

  1. ''FN-Au and Col-Au have a significantly lower inflammatory response, plate- 914 let activation degree, apoptotic rate, and ROS generation'' is this compared to what?

Response:

Thanks for the valuable comment from the reviewer. We have revised the “Conclusion” section and included the description to “compare to pure FN and Col which refer to” (Line 931)

  1. For the ORO and ARS staining, positive controls should be included to compare the figures with the appearance of true positives.

Response:

MSCs inherently have the potential property for multiple self-differentiations. Therefore, stem cells in the control group will also express ORO and ARS intensity. At the same time, the quantitative data of the research results showed that FN-Au and Col-Au promoted ORO and ARS significantly higher than the control group.

Comments on the Quality of English Language

Extensive revision of the writing is needed. It seems as the authors didn't take time to carefully prepare the manuscript's method part, which appears it has been simply copied and pasted from an internal protocol.

Response:

Thanks for the valuable comments from the reviewer. We have approved this revision to external English editing company for this manuscription.

We also find a wrongly representation in the “Figure 1B”. We have replaced the new image in the Figure 1B and modified the description in the

  • “Results” section “roughness (Ra) of FN, FN-Au, Col, and Col-Au is4 nm, 1.04 nm, 0.97 nm, and 1.66 nm” (Line 193, line 195-196)
  • In the “Discussion” section “4 nm to 1.04 nm” and “(1.66 nm) than Col (0.97 nm)” (Line 537, line 539)

Round 2

Reviewer 2 Report

Comments and Suggestions for Authors

All the figures  including in the supplementary part, must include scale bars. All the scale bars should further labelled with the corresponding length (or include this information in the legend).

Check the syntax and writing of the all manuscript. Numerous sentences are incorrectly written. In the methods you need to written as for De Mori et al. 2024,  ovine mesenchymal stem cells etc. For instance, line 955, "prepare a 2%...:. , 942, 943 etc.

Comments on the Quality of English Language

Extensive revision of manuscript for the syntax is needed.

Author Response

It is very much appreciated that the reviewer made these new valuable comments. We have attended to the comments and made the revision accordingly.

Response to Reviewer 2 Comments:

Comments and Suggestions for Authors

1. All the figures including in the supplementary part, must include scale bars. All the scale bars should further labelled with the corresponding length (or include this information in the legend).

Response:

Thanks for the valuable new comment. We have included the error bar in each figure and described in the “Figure caption” section. 

2. Check the syntax and writing of the all manuscript. Numerous sentences are incorrectly written. In the methods you need to written as for De Mori et al. 2024, ovine mesenchymal stem cells etc. For instance, line 955, "prepare a 2%...:. , 942, 943 etc.

Response:

1. Thanks for the valuable comment. We cannot to know and to find what’s mean about “e Mori et al. 2024, ovine mesenchymal stem cells etc. For instance, line 955, "prepare a 2%...:. , 942, 943 etc.” after carefully go through for this manuscript (original R1: Line 942-943, Line 955) following by Reviewer’s suggestion.

2. Thanks for the valuable comment. We have approval for this manuscript to external English editing company in the second revision (please see attach file).

Comments on the Quality of English Language

Extensive revision of manuscript for the syntax is needed.

Response:

Thanks for the valuable comment from the reviewer. We have approval this manuscript to external English editing company following your kindness suggestion for the round 1 (2024/06/09) and round 2 revision process (2024/06/14) (please see attach file).
